# Revisiting Instance-Optimal Cluster Recovery
# in the Labeled Stochastic Block Model

**Kaito Ariu** [1] **Alexandre Proutiere** [2] **Se-Young Yun** [3]

## Abstract

In this paper, we investigate the problem of recovering hidden communities in the Labeled Stochastic Block Model (LSBM) with a finite number of clusters whose sizes grow linearly with the total number of nodes. We derive the necessary and sufficient conditions under which the expected number of misclassified nodes is less than $s$, for any number $s = o(n)$. To achieve this, we propose IAC (Instance-Adaptive Clustering), the first algorithm whose performance matches the instance-specific lower bounds both in expectation and with high probability. IAC is a novel two-phase algorithm that consists of a one-shot spectral clustering step followed by iterative likelihood-based cluster assignment improvements. This approach is based on the instance-specific lower bound and notably does not require any knowledge of the model parameters, including the number of clusters. By performing the spectral clustering only once, IAC maintains an overall computational complexity of $\mathcal{O}(n \operatorname{polylog}(n))$, making it scalable and practical for large-scale problems.

## 1. Introduction

Community detection or clustering refers to the task of gathering similar nodes into a few groups from the data that, most often, correspond to observations of pair-wise interactions between nodes (Newman & Girvan, 2004). A benchmark commonly used to assess the performance of clustering algorithms is the celebrated Stochastic Block Model (SBM) (Holland et al., 1983), where pair-wise interactions are represented by a random graph. In this graph, the vertices correspond to nodes, and the presence of an edge between two nodes indicates their interaction.

The SBM has been widely studied over the past two decades;

[1]CyberAgent [2]KTH, Digital Futures [3]KAIST. Correspondence to: Kaito Ariu <kaito_ariu@cyberagent.co.jp>.

*Proceedings of the 42$^{nd}$ International Conference on Machine Learning*, Vancouver, Canada. PMLR 267, 2025. Copyright 2025 by the author(s).

for a detailed overview, see (Abbe, 2018). However, it offers a somewhat simplistic representation of how nodes interact. In real-world applications, interactions can vary in type—such as ratings in recommender systems or proximity levels in social networks. To capture this richer, more nuanced interaction data, the Labeled Stochastic Block Model (LSBM) was introduced and analyzed in (Heimlicher et al., 2012; Lelarge et al., 2013; Yun & Proutiere, 2016). LSBM represents interactions through labels drawn from an arbitrary set. The aim of this paper is to illustrate the tightest condition under which clustering is possible based on observing these labels. For that purpose, we develop a clustering algorithm that minimizes the expected number of misclassified nodes. In the following sections, we formally introduce LSBMs and summarize our results.

**The Labeled Stochastic Block Model.** In the LSBM, the set $\mathcal{I}$ consisting of $n$ items or nodes is randomly partitioned into $K$ unknown disjoint clusters $\mathcal{I}_1, \ldots, \mathcal{I}_K$. The cluster index of the node $i$ is denoted by $\sigma(i)$. Let $\alpha = (\alpha_1, \alpha_2, \ldots, \alpha_K)$ represent the probabilities of nodes belonging to each cluster, i.e., for all $k \in [K]$ and $i \in \mathcal{I}$, $\mathbb{P}(i \in \mathcal{I}_k) = \alpha_k$. We assume that $\alpha_1, \ldots, \alpha_K$ are strictly positive constants and that $K$ and $\alpha$ are fixed as $n$ grows large. The number of clusters $K$ is initially unknown. Without loss of generality, we also assume that $\alpha_1 \leq \ldots \leq \alpha_K$. Let $\mathcal{L} = \{0, 1, \ldots, L\}$ be the finite set of labels. For each edge $(v, w) \in \mathcal{I}_i \times \mathcal{I}_j$, the learner observes the label $\ell$ with probability $p(i, j, \ell)$, independently of the labels observed in other edges. We have $\forall i, j \in [K]^2$, $\sum_{\ell \in \mathcal{L}} p(i, j, \ell) = 1$. Without loss of generality, $0$ is the most frequent label: $0 = \arg\max_\ell \sum_{i=1}^{K} \sum_{j=1}^{K} \alpha_i \alpha_j p(i, j, \ell)$. Let $\bar{p} = \max_{i,j,\ell \geq 1} p(i, j, \ell)$ be the maximum probability of observing a label different from $0$. We will mostly consider the challenging sparse regime where $\bar{p} = \mathcal{O}((\log n)/n)$ and $\bar{p}n \to \infty$ as $n \to \infty$, but we will precise the assumptions made on $n$ and $\bar{p}$ for each of our results. We further assume for all $i, j, k \in [K]$:

(A1) $\forall \ell \in \mathcal{L}$, $\dfrac{p(i, j, \ell)}{p(i, k, \ell)} \leq \eta$ and

(A2) $\dfrac{\sum_{k=1}^{K} \sum_{\ell=1}^{L} (p(i, k, \ell) - p(j, k, \ell))^2}{\bar{p}^2} \geq \varepsilon$,

where $\eta$ and $\varepsilon$ are positive constants independent of $n$. (A1)

imposes some homogeneity on the edge existence probability, and (A2) implies a certain separation among the clusters. In summary, the LSBM is parametrized by $\alpha$ and $p \coloneqq (p(i,j,\ell))_{1 \leq i,j \leq K, 0 \leq \ell \leq L}$.

## 1.1. Notation

We denote $p(i)$ as the $K \times (L+1)$ matrix whose element on $j$-th row and $(\ell+1)$-th column is $p(i,j,\ell)$ and denote $p(i,j) \in [0,1]^{L+1}$ the vector describing the probability of the label of a pair of nodes in $\mathcal{I}_i$ and $\mathcal{I}_j$. Let $\mathcal{P}^{K \times (L+1)}$ denote the set of all $K \times (L+1)$ matrices such that each row represents a probability distribution. Define the divergence $D(\alpha, p)$ of the parameter $(\alpha, p)$ as: $D(\alpha, p) = \min_{i,j \in [K]: i \neq j} D_{L+}(\alpha, p(i), p(j))$, with

$$D_{L+}(\alpha, p(i), p(j)) = \min_{y \in \mathcal{P}^{K \times (L+1)}} \max$$

$$\left\{ \sum_{k=1}^{K} \alpha_k \, \mathrm{kl}(y(k), p(i,k)), \sum_{k=1}^{K} \alpha_k \, \mathrm{kl}(y(k), p(j,k)) \right\},$$

and where kl denotes the Kullback-Leibler divergence between two label distributions, i.e., $\mathrm{kl}(y(k), p(i,k)) = \sum_{\ell=0}^{L} y(k,\ell) \log \frac{y(k,\ell)}{p(i,k,\ell)}$. $D_{L+}(\alpha, p(i), p(j))$ can be interpreted as the hardness in distinguishing whether a node belongs to cluster $i$ or cluster $j$ based on the data. For any clustering algorithm $\pi$, we denote by $\varepsilon^{\pi}(n)$ the number of misclassified nodes under $\pi$. This quantity is defined up to a permutation of the cluster IDs. Specifically, if $\pi$ returns $(\hat{\mathcal{I}}_k)_k$, then $\varepsilon^{\pi}(n)$ is calculated as $\min_{\theta} | \cup_k \hat{\mathcal{I}}_k \setminus \mathcal{I}_{\theta(k)}|$, where the minimum is over all permutations $\theta$ of $[K]$. To simplify the notation throughout the paper, we assume that the permutation achieving the minimum is given by $\theta(k) = k$ for all $k \in [K]$.

## 1.2. Main Results

We consider the problem of clustering, or recovering communities, in the LSBM as described above. We reveal the necessary and sufficient conditions for the asymptotic recovery of the communities. Specifically, we reveal that the necessary and sufficient condition on the parameters $(\alpha, p)$ to achieve $\limsup_{n \to \infty} \frac{\mathbb{E}[\varepsilon^{\pi}(n)]}{s} \leq 1$, for any $s = o(n)$, is

$$\liminf_{n \to \infty} \frac{nD(\alpha, p)}{\log(n/s)} \geq 1.$$

To this aim, we design a computationally efficient algorithm that recovers the clusters in the LSBM with a minimal error rate. By *minimal*, we mean that for any given LSBM, the algorithm achieves the best possible error rate for this specific LSBM. In other words, the algorithm is instance-optimal and truly adapts to the hardness of the LSBM it faces. This contrasts with algorithms with minimax performance guarantees only: those algorithms provably perform well for the worst possible LSBM, but could potentially yield poor performance for most other instances of the LSBM.

**Instance-specific lower bound.** To reveal the necessary condition, we first need to revisit an instance-specific lower bound on the error rate satisfied by any algorithm. The following theorem from (Yun & Proutiere, 2016) provides a necessary condition as a lower bound on the expected number of misclassified nodes $\mathbb{E}[\varepsilon^{\pi}(n)]$.

**Theorem 1.1** (Yun & Proutiere (2016))**.** *Let* $s = o(n)$*. Under the assumptions of (A1), (A2), and* $\bar{p}n = \omega(1)$*, for any clustering algorithm* $\pi$ *that satisfies* $\limsup_{n \to \infty} \frac{\mathbb{E}[\varepsilon^{\pi}(n)]}{s} \leq 1$*,*

$$\liminf_{n \to \infty} \frac{nD(\alpha, p)}{\log(n/s)} \geq 1.$$

The lower bound essentially states that under any algorithm, the expected number of misclassified nodes must be larger than $n \exp(-nD(\alpha, p))$. The proof of Theorem 1.1 is based on the change-of-measure argument frequently used in online stochastic optimization and multi-armed bandit problems (Kaufmann et al., 2016; Lai & Robbins, 1985).

**An instance-optimal algorithm (upper bound).** The main contribution of this paper is to provide a sufficient condition for cluster recovery. Specifically, we propose an algorithm with performance guarantees that match those of the above lower bound and with computational complexity scaling as $n\mathrm{polylog}(n)$. This algorithm, referred to as Instance-Adaptive Clustering (IAC) and presented in Section 3, first applies a spectral clustering algorithm to initially guess the clusters and then runs a *likelihood-based local improvement* algorithm to refine the estimated clusters. To analyze the performance of the algorithm, we make the following assumption.

(A3)    $np(j,i,\ell) \geq (n\bar{p})^{\kappa}$ for all $i,j$ and $\ell \geq 1$,

       for some constant $\kappa > 0$.

Assumption (A3) excludes the existence of labels that are too sparse compared to $\bar{p}$. The following theorem establishes the performance guarantee of IAC. Its full proof is given in Appendix C.

**Theorem 1.2.** *Assume that (A1), (A2), and (A3) hold, and that* $\bar{p} = O(\log n/n)$*,* $\bar{p}n = \omega(1)$*. Let* $s = o(n)$*. If the parameters* $(\alpha, p)$ *of the LSBM satisfy*

$$\liminf_{n \to \infty} \frac{nD(\alpha, p)}{\log(n/s)} \geq 1. \tag{1}$$

*then IAC (Algorithm 1) misclassifies at most* $s$ *nodes in high probability and in expectation, i.e.,*

$$\lim_{n \to \infty} \mathbb{P}[\varepsilon^{\mathrm{IAC}}(n) \leq s] = 1 \text{ and } \limsup_{n \to \infty} \frac{\mathbb{E}[\varepsilon^{\mathrm{IAC}}(n)]}{s} \leq 1.$$

*IAC requires $\mathcal{O}(n(\log n)^3)$ floating-point operations.*

As far as we are aware, IAC is the first algorithm achieving a performance that matches the lower bound presented in Theorem 1.1.

To obtain an instance-optimal guarantee in expectation, we had to (i) reshape parts of the algorithm, and (ii) develop new tools to analyze its performance. Specifically, to assess the expected number of misclassified nodes after the first spectral clustering phase, we needed a tighter guarantee for the probability of a failure event of the spectral clustering algorithm with the iterative power method and singular value thresholding. Analyzing the second phase (the likelihood-based improvement step) of the algorithm in expectation was also challenging. All intermediate statements had to hold with sufficiently high probability and quantify the number of misclassified nodes for any number $s = o(n)$. Furthermore, as reflected in the definition of the divergence $D(\alpha, p)$, the optimal clustering boundary is asymmetric, necessitating that both the improvement step and its analysis account for the nonlinear information geometry.

## 2. Related Work

In this section, we review existing literature on community detection in stochastic block models (SBMs) and their extensions, focusing on performance guarantees and optimal recovery rates relevant to our analysis.

### 2.1. Community Detection in the SBM

Community detection in the stochastic block model (SBM) and its extensions has received considerable attention over the last decade. We first briefly outline existing results below, and then focus on a few papers that are most relevant to our analysis. The results for the SBM can be categorized according to the type of performance guarantee targeted. We distinguish three types of guarantees: detectability, asymptotically accurate recovery, and exact recovery. Most results are concerned with the standard SBM, which can be obtained as a special case of the LSBM (with $L = 1$, where the intra- and inter-cluster probabilities are denoted by $p(i, i, 1)$ and $p(i, j, 1)$ for $i \neq j \in [K]$).

**Detectability.** Detectability refers to the requirement of returning clusters that are positively correlated with the true clusters. It is typically studied in the sparse binary SBM where $K = 2$, $\alpha_1 = \alpha_2$, $p(1, 1, 1) = p(2, 2, 1) = a/n$ and $p(1, 2, 1) = p(2, 1, 1) = b/n$, for some constants $a > b$ independent of $n$. For such SBM, detectability can be achieved if and only if $(a - b) > \sqrt{2(a + b)}$ (Decelle et al., 2011; Mossel et al., 2015a; Massoulié, 2013). Detectability conditions in more general sparse SBMs have been investigated in (Krzakala et al., 2013; Bordenave et al., 2015). In the sparse SBM, when the edge probabilities scale as

$\mathcal{O}(1/n)$, there is a positive fraction of isolated nodes, and we cannot do much better than merely detecting the clusters. In this paper, we focus on scenarios where the edge probabilities are $\omega(1/n)$, where an asymptotically accurate recovery of the clusters may be achieved.

**Asymptotically accurate recovery.** Our results can also be interpreted as a generalization of the necessary and sufficient condition for asymptotically accurate recovery of the communities in the labeled stochastic block model (LSBM), namely $nD(\alpha, p) = \omega(1)$. Under this condition, we can achieve asymptotically accurate recovery of the clusters, meaning that the proportion of misclassified nodes tends to zero as $n$ grows large. A necessary and sufficient condition for asymptotically accurate recovery in the SBM (with any number of clusters of different but linearly increasing sizes) has been derived in (Yun & Proutiere, 2014b) and (Mossel et al., 2015b). In our work, we performed a more precise analysis and succeeded in generalizing this condition. Our results are characterized by our divergence $D(\alpha, p)$, which makes the condition *instance-specific*. Our analysis thus provides more accurate results than those derived in a minimax framework (Gao et al., 2017; Xu et al., 2020; Fei & Chen, 2020; Zhang & Zhou, 2020). An extensive comparison with (Gao et al., 2017), where we also strengthen their analysis, is provided in Section 2.2.

**Asymptotically exact recovery.** An algorithm achieves an asymptotically exact recovery if no nodes are misclassified asymptotically. By choosing $s = 1/2$ in our theorems, our results provide the necessary and sufficient condition for such exact recovery. Conditions for asymptotically exact recovery have also been studied in the binary symmetric SBM (Yun & Proutiere, 2014a; Abbe et al., 2016; Mossel et al., 2015b; Hajek et al., 2016). For instance, when $K = 2$, $\alpha_1 = \alpha_2$, $p(1, 1, 1) = p(2, 2, 1) = \frac{a \log n}{n}$ and $p(1, 2, 1) = p(2, 1, 1) = \frac{b \log n}{n}$, where $a$ and $b$ are some constants with $a > b$, cluster recovery is possible if and only if $\frac{a+b}{2} - \sqrt{ab} \geq 1$. There are also extensions to more general SBMs (Abbe & Sandon, 2015a;b; Wang et al., 2021). Our results are in a form that generalizes these results.

### 2.2. Comparison with Previous Work on Optimal Recovery Rates

Next, we discuss three papers (Zhang & Zhou, 2016; Gao et al., 2017; Xu et al., 2020) that are directly related to our analysis. These papers study the standard SBM or homogeneous LSBM in the regime where asymptotically accurate recovery is possible. (Zhang & Zhou, 2016; Gao et al., 2017) present the minimal expected number of misclassified nodes, but in a minimax setting, whereas (Xu et al., 2020) presents the instance-specific analysis.

The authors of (Zhang & Zhou, 2016) characterize the min-

imal expected number of misclassified nodes in the *worst* possible SBM within the class $\Theta(n, a, b)$ of SBMs satisfying, using our notation, $p(i, i, 1) \geq \frac{a}{n}$ and $p(i, j, 1) \leq \frac{b}{n}$ for all $i \neq j \in [K]$, where $a$ and $b$ are positive constants. To simplify the exposition here, we assume that all clusters are of equal size (refer to (Zhang & Zhou, 2016) for more details). The minimal expected number of misclassified nodes is defined through the Rényi divergence of order $\frac{1}{2}$ between the Bernoulli random variables of respective means $\frac{a}{n}$ and $\frac{b}{n}$, given by

$$I^*(n, a, b) = -2\log\left(\sqrt{\frac{a}{n}}\sqrt{\frac{b}{n}} + \sqrt{1 - \frac{a}{n}}\sqrt{1 - \frac{b}{n}}\right).$$

When $nI^*(n, a, b) = \omega(1)$, $\mathbb{E}[\varepsilon^\pi(n)]$ scales as $n\exp(-(1 + o(1))\frac{nI^*(n,a,b)}{K})$. (Zhang & Zhou, 2016) established that the so-called penalized Maximum Likelihood Estimator (MLE) achieves this minimax optimal recovery rate but does not provide any algorithm to compute it.

The authors of (Gao et al., 2017) present an algorithm that runs in polynomial time and achieves this minimax lower bound with high probability[1]. Again, for simplicity, we assume that all clusters are of equal size and that $K$ is a fixed constant. Their performance guarantee can be stated as follows (see Theorem 4 in (Gao et al., 2017)):

$$\sup_{(\alpha,p)\in\Theta(n,a,b)} \mathbb{P}_{(\alpha,p)}$$
$$\left(\varepsilon^\pi(n) \geq n\exp\left(-(1 + o(1))\frac{nI^*(n,a,b)}{K}\right)\right) \to 0,$$

where $\mathbb{P}_{(\alpha,p)}$ denotes the distribution of the observations generated under the SBM $(\alpha, p)$. One could argue that the above guarantee does not match the minimax lower bound valid for the *expected* number of misclassified nodes. However, by carefully inspecting the proof of Theorem 4 in (Gao et al., 2017), it is easy to see that the guarantee also holds in expectation:

**Corollary 2.1.** *Assume that $a/b = \Theta(1)$, $(a - b)^2/a = \omega(1)$, and $a = \mathcal{O}(\log(n))$. Let $A_{uv}$ be the observation for the pair of nodes $(u, v)$. Under Algorithm 1 in (Gao et al., 2017) initialized with $\text{USC}(\tau)$ in (Gao et al., 2017) for $\tau = C\frac{1}{n}\sum_{u\in[n]}\sum_{v\in[n]} A_{uv}$ with some large enough constant $C > 0$,*

$$\sup_{(\alpha,p)\in\Theta(n,a,b)} \mathbb{E}_{(\alpha,p)}[\varepsilon^\pi(n)]$$
$$\leq n\exp\left(-(1 + o(1))\frac{nI^*(n,a,b)}{K}\right).$$

The proof is presented in Appendix E.10. The assumptions of Corollary 2.1 are satisfied when our assumptions (A1) and (A2) hold.

The algorithm presented in (Gao et al., 2017), which has established performance guarantees, comes with a high computational cost. It requires applying spectral clustering $n$ times, where for each node $u$, the algorithm builds a modified adjacency matrix by removing the $u$-th column and the $u$-th row and then computes a spectral clustering of this matrix. In contrast, our algorithm performs spectral clustering only once. (Gao et al., 2017) also proposed an algorithm with reduced complexity (running in $\Omega(n^2)$), but without performance guarantees. Our algorithm not only performs spectral clustering once but also requires just $\mathcal{O}(n(\log n)^3)$ operations. Additionally, our algorithm empirically exhibits better classification accuracy than the penalized local maximum likelihood estimation algorithm from (Gao et al., 2017) in several simple scenarios (see Appendix F).

To conclude, compared to (Gao et al., 2017), our analysis provides an instance-specific lower bound for the classification error probability (rather than minimax) and introduces a low-complexity algorithm that matches this lower bound. Additionally, our analysis is applicable to the generic Labeled SBMs. It is worth noting, however, that (Gao et al., 2017) derives upper bounds for classification error probability under slightly more general assumptions than ours for the SBMs. Specifically, their high probability guarantee also holds for dense regimes ($\bar{p} = \omega(\log n/n)$), and their number of clusters $K$ can depend on $n$, whereas we consider fixed $K$.

(Xu et al., 2020) presents an instance-specific analysis for the homogeneous LSBM, in which the distribution of an edge depends solely on whether the corresponding nodes are in the same cluster or not. Specifically, there exist distributions $P$ and $Q$ such that for each edge label $\ell$, the probabilities are defined as $p(i, i, \ell) = P(\ell)$ for all $i \in [K]$, and $p(i, j, \ell) = Q(\ell)$ for all $i \neq j$. In contrast, our model does not impose such restrictions; the edge label probabilities can vary heterogeneously depending on the cluster indices. Similar to the algorithm in (Gao et al., 2017), the algorithm proposed in (Xu et al., 2020) executes spectral clustering $n$ times (see lines 8–10 in Algorithm 3 of (Xu et al., 2020)).

### 2.3. Other Related Work

**Spectral methods.** Recent studies also suggest that there are clustering instances where the spectral clustering algorithm alone is optimal without the improvement step. (Zhang, 2024) provides a detailed analysis of the spectral method without any improvement steps in the setting of simple SBMs for asymptotically accurate recovery. Their work establishes lower bounds on the performance achievable by spectral clustering algorithms. However, these lower

---

[1]Refer to (Gao et al., 2017) for the class of SBMs considered. Compared to our assumptions (A1)-(A2)-(A3) specialized to SBMs, this class of SBMs is slightly more general.

bounds are specific to the spectral method and thus do not represent the fundamental limits of community detection that we have been considering. As a result, the focus of their work is not on developing algorithms that achieve optimal rates but rather on providing a tight analysis of the spectral method's performance. We also conjecture that in the general (L)SBM, spectral clustering alone cannot achieve the optimal rates due to the asymmetry of the optimal decision boundary, as reflected in the definition of the divergence $D(\alpha, p)$. This necessitates the use of non-Euclidean distances within the algorithm. In (Zhang, 2024), the $k$-means algorithm is based on the Euclidean distance (see Eq. (2) of Algorithm 1 in (Zhang, 2024)), which would be insufficient to achieve our optimal result.

For the exact recovery regime, (Dhara et al., 2024; Gaudio & Liu, 2024) discuss the optimality of the spectral algorithm for the Censored SBM (which is an instance of LSBM) or LSBM. They only provided high-probability guarantees and the algorithms require weights that depend on the statistical parameters.

**Related models.** LSBM generalizes the SBM, the Censored SBM (Abbe et al., 2014; Dhara et al., 2024), and signed networks (Traag & Bruggeman, 2009; Leskovec et al., 2010; Tzeng et al., 2020), and has many applications. Our results provide necessary and sufficient conditions for cluster recovery in all of these models for any $s = o(n)$. The Multiplex Stochastic Block Model (Barbillon et al., 2016) and the node-attributed Stochastic Block Model (Dreveton et al., 2023) are interesting because they can capture cluster structures that cannot be represented by the SBM or the LSBM. Extending our work to such models is an important future direction.

## 3. Instance-Adaptive Clustering Algorithm

In this section, we introduce the *Instance-Adaptive Clustering* (IAC) algorithm, outlined in Algorithm 1. The IAC algorithm is a two-step algorithm consisting of initial spectral clustering and iterative likelihood-based improvements. The algorithm aims to improve clustering performance by adapting to the instances.

### 3.1. Algorithms

The Instance-Adaptive Clustering (IAC), whose pseudocode is presented in Algorithm 1, consists of two phases: a spectral clustering initialization phase and a likelihood-based improvement phase.

**Spectral clustering initialization.** The algorithm relies on simple spectral techniques to obtain rough estimates of the clusters. For details, refer to lines 1-4 in Algorithm 1 and Algorithm 2. The algorithm first constructs an observation matrix $A^\ell = (A^\ell_{uv})_{u,v}$ for each label $\ell$ (where $A^\ell_{uv} = 1$

---

**Algorithm 1** Instance-Adaptive Clustering

**Input:** Observed adjacency matrices $(A^\ell)_{\ell \in [L]}$
   (where $A^\ell_{uv} = 1$ if $\ell$ is observed between $u$ and $v$)
1: **1. Estimate average degree:**
2: $\widetilde{p} \leftarrow (\sum_{\ell=1}^{L} \sum_{v,w \in \mathcal{I}: v > w} A^\ell_{vw})/n(n-1)$
3: **2. Trimming:**
4: Compute $(A^\ell_\Gamma)_{\ell \in [L]}$, where $\Gamma$ is the set of remaining nodes after removing the nodes with the top $\lfloor n \exp(-n\widetilde{p}) \rfloor$ largest values of $\sum_{\ell=1}^{L} \sum_{w \in \mathcal{I}} A^\ell_{vw}$
5: For all $\ell \in [L]$ and $w, v \in \mathcal{I}$:
6: $\quad (A^\ell_\Gamma)_{wv} \leftarrow \begin{cases} A^\ell_{wv}, & \text{if } w, v \in \Gamma \\ 0, & \text{if } w, v \in \Gamma^c \end{cases}$
7: **3. Spectral Clustering:**
8: Run Algorithm 2 with input $(A^\ell_\Gamma)_{\ell \in [L]}$, $\widetilde{p}$ and output $\{S_k\}_{k=1}^{\hat{K}}$
9: **4. Estimation of the Statistical Parameters:**
10: **for** each $i, j \in [\hat{K}]$ and $\ell \in \{0, 1, \ldots, L\}$ **do**
11: $\quad \hat{p}(i, j, \ell) \leftarrow (\sum_{u \in S_i} \sum_{v \in S_j} A^\ell_{uv})/|S_i||S_j|$
12: **end for**
13: **5. Likelihood-based local improvements:**
14: $S_k^{(0)} \leftarrow S_k$ for all $k \in [\hat{K}]$
15: **for** $t = 1$ **to** $\log n$ **do**
16: $\quad$ **for** each $k \in [\hat{K}]$ **do**
17: $\quad\quad S_k^{(t)} \leftarrow \emptyset$
18: $\quad$ **end for**
19: $\quad$ **for** each $v \in \mathcal{I}$ **do**
20: $\quad\quad k^* \leftarrow \arg\max\{\sum_{1 \le k \le \hat{K}} \sum_{i \in [\hat{K}]} \sum_{w \in S_i^{(t-1)}} \sum_{\ell=0}^{L} A^\ell_{vw}$
   $\log \hat{p}(k, i, \ell)\}$ (tie broken uniformly at random)
21: $\quad\quad S_{k^*}^{(t)} \leftarrow S_{k^*}^{(t)} \cup \{v\}$
22: $\quad$ **end for**
23: **end for**
24: $\hat{\mathcal{I}}_k \leftarrow S_k^{(\log n)}$ for all $k \in [\hat{K}]$
**Output:** Clusters $(\hat{\mathcal{I}}_k)_{k=1,\ldots,\hat{K}}$

---

iff label $\ell$ is observed on edge $(u, v)$). After the trimming process, which eliminates rows and columns corresponding to nodes with an excessive number of observed labels (as these could distort the spectral properties of $A^\ell$ as $A^\ell_\Gamma$, we apply spectral clustering to $(A^\ell_\Gamma)_{\ell \in [L]}$. Specifically, we concatenate the matrices $(A^\ell_\Gamma)_{\ell \in [L]}$ as $\tilde{A} = [A^1_\Gamma, A^2_\Gamma, \ldots, A^L_\Gamma]$ and use the iterative power method (instead of using a direct SVD) combined with singular value thresholding (Chatterjee, 2015). This approach allows us to control the computational complexity of the algorithm and to accurately estimate the number of clusters.

Notable differences compared to the spectral clustering phase of the algorithm presented in (Yun & Proutiere, 2016) include (i) a change in the number of matrix multiplications required in the iterative power method, which is now ap-

proximately $(\log n)^2$; (ii) an expansion of the set of centroid candidates in the k-means algorithm, which now includes $(\log n)^2$ randomly selected nodes; and (iii) the use of the concatenated matrix $\bar{A} = [A_\Gamma^1, A_\Gamma^2, \ldots, A_\Gamma^L]$. In contrast, (Yun & Proutiere, 2016) uses a randomly weighted matrix $\sum_{\ell \in [L]} w_\ell A_\Gamma^\ell$ with i.i.d. uniform distribution $w_\ell$ for clustering, which prevents us from obtaining the guarantee in expectation. Our new approach allows for tighter control of the probability of failure events and provides guarantees in expectation.

**Likelihood-based improvements.** Using the initial cluster estimates $S_i$, we can also estimate $p$ from the data. For any $i, j, \ell$, we calculate the empirical estimate:

$$\hat{p}(i, j, \ell) = \frac{\sum_{u \in S_i} \sum_{v \in S_j} A_{uv}^\ell}{|S_i||S_j|}.$$

Based on $\hat{p}$, the log-likelihood of node $v$ belonging to cluster $S_k$ is computed as $\sum_{i \in [\hat{K}]} \sum_{w \in S_i} \sum_{\ell=0}^{L} A_{vw}^\ell \log \hat{p}(k, i, \ell)$. Subsequently, $v$ is assigned to the cluster that maximizes this log-likelihood over $[\hat{K}]$. This process is applied to all nodes and iterated $\log n$ times. One iteration of this process is fast: by taking advantage of sparsity and computing by traversing only the node pairs for which a label exists, the computational complexity can be reduced to $\mathcal{O}(\log n)$. The improvement step corresponds to improving the cluster assignment along the nonlinear decision boundary, as reflected in the definition of our divergence $D(\alpha, p)$.

# 4. Theoretical Guarantees

In this section, we provide a sketch of the proof of its performance guarantees, with the full proof available in Appendix C.2.

## 4.1. Spectral Clustering Initialization

The following theorem establishes performance guarantees for the cluster estimates returned by the spectral clustering algorithm (refer to Algorithm 2 for details). Specifically, we show that the number of clusters is correctly predicted as $\hat{K} = K$, and the number of misclassified nodes is $\mathcal{O}(1/\bar{p})$, which is $o(n)$ as $n\bar{p} = \omega(1)$.

**Theorem 4.1.** *Assume that (A1) and (A2) hold. After Algorithm 2, for any $c > 0$, there exists a constant $C > 0$ such that*

$$\hat{K} = K \quad and \quad \min_\theta \left| \bigcup_{k=1}^{K} S_k \setminus \mathcal{I}_{\theta(k)} \right| \le \frac{C}{\bar{p}},$$

*with probability at least $1 - \exp(-cn\bar{p})$, where the minimization is performed over the permutation $\theta$ of $[K]$.*

*Proof Sketch of Theorem 4.1.* Let $M^\ell$ denote the expectation of the matrix $A^\ell$: $M_{uv}^\ell = p(i, j, \ell)$ when $u \in \mathcal{V}_i$

and $v \in \mathcal{V}_j$. Let $M = [M^1, M^2, \ldots, M^L]$, and $M_\Gamma = [M_\Gamma^1, M_\Gamma^2, \ldots, M_\Gamma^L] \in [0, 1]^{n \times Ln}$ be the corresponding trimmed matrix, where $(M_\Gamma^\ell)_{wv} = M_{wv}^\ell \mathbb{1}_{\{w, v \in \Gamma\}}$.

---

**Algorithm 2** Spectral Clustering

**Input:** $(A_\Gamma^\ell)_{\ell \in [L]}$, $\widetilde{p}$
1: **1. Concatenation:**
2: $\quad \bar{A} \leftarrow [A_\Gamma^1, A_\Gamma^2, \ldots, A_\Gamma^L]$
3: **2. Iterative Power Method with Singular Value Thresholding:** $\chi \leftarrow n, \quad k \leftarrow 0, \quad \hat{U} \leftarrow \mathbf{0}^{nL \times 1}$

4: **while** $\chi \ge \sqrt{n\widetilde{p}} \log(n\widetilde{p})$ **do**
5: $\quad k \leftarrow k + 1$
6: $\quad U_0 \leftarrow$ random Gaussian vector of size $nL \times 1$
7: $\quad$ *(Iterative power method)* $U_t \leftarrow (\bar{A})^{2\lceil (\log n)^2 \rceil + 1} U_0$
8: $\quad$ *(Orthonormalizing)* $\hat{U}_k \leftarrow$
9: $\quad U_t - \hat{U}_{1:k-1}(\hat{U}_{1:k-1}^\top U_t) / \left\| U_t - \hat{U}_{1:k-1}(\hat{U}_{1:k-1}^\top U_t) \right\|_2$
10: $\quad$ *(Estimated k-th singular value)* $\chi \leftarrow \left\| \bar{A}\hat{U}_k \right\|_2$
11: **end while**
12: $\hat{K} \leftarrow k - 1, \quad \hat{A} \leftarrow \hat{U}_{1:\hat{K}}^\top \bar{A}$
13: **3. k-means Clustering:**
14: $\mathcal{I}_R \leftarrow$ a subset of $\Gamma$ obtained by randomly selecting $\lceil (\log n)^2 \rceil$ nodes
15: **for** $t = 1$ **to** $\lceil \log n \rceil$ **do**
16: $\quad$ **for** each $v \in \mathcal{I}_R$ **do**
17: $\quad\quad Q_v^{(t)} \leftarrow \left\{ w \in \mathcal{I} \mid \left\| \hat{A}_w - \hat{A}_v \right\|_2^2 \le t\frac{\widetilde{p}}{100} \right\}$
18: $\quad$ **end for**
19: $\quad T_k^{(t)} \leftarrow \emptyset$ for all $k \in [\hat{K}]$
20: $\quad$ **for** $k = 1$ **to** $\hat{K}$ **do**
21: $\quad\quad v_k^* \leftarrow \arg\max_{v \in \mathcal{I}_R} \left| Q_v^{(t)} \setminus \bigcup_{i=1}^{k-1} T_i^{(t)} \right|$
22: $\quad\quad T_k^{(t)} \leftarrow Q_{v_k^*}^{(t)} \setminus \bigcup_{i=1}^{k-1} T_i^{(t)}$
23: $\quad\quad \xi_k^{(t)} \leftarrow \frac{1}{\left| T_k^{(t)} \right|} \sum_{v \in T_k^{(t)}} \hat{A}_v$
24: $\quad$ **end for**
25: $\quad$ **for** each $v \in \mathcal{I} \setminus \bigcup_{k=1}^{\hat{K}} T_k^{(t)}$ **do**
26: $\quad\quad k^* \leftarrow \arg\min_{1 \le k \le \hat{K}} \left\| \hat{A}_v - \xi_k^{(t)} \right\|_2^2$
27: $\quad\quad T_{k^*}^{(t)} \leftarrow T_{k^*}^{(t)} \cup \{v\}$
28: $\quad$ **end for**
29: $\quad r_t \leftarrow \sum_{k=1}^{\hat{K}} \sum_{v \in T_k^{(t)}} \left\| \hat{A}_v - \xi_k^{(t)} \right\|_2^2$
30: **end for**
31: $t^* \leftarrow \arg\min_{1 \le t \le \lceil \log n \rceil} r_t$
32: $S_k \leftarrow T_k^{(t^*)}$ for all $k \in [\hat{K}]$
**Output:** Clusters $\{S_k\}_{k=1}^{\hat{K}}$

---

(a) The first ingredient of the proof is to use an upper bound

on the norm of the noise matrix $X_\Gamma^\ell = A_\Gamma^\ell - M_\Gamma^\ell$, which holds with sufficiently high probability, as stated in the following lemma from (Le et al., 2017, Theorem 2.1).

**Lemma 4.2.** *For any $\ell \in [L]$, for any $C > 0$, there exists $C' > 0$ such that:*

$$||X_\Gamma^\ell||_2 \leq C'\sqrt{n\bar{p}} ,$$

*with probability at least $1 - \exp(-Cn\bar{p})$.*

We note that such a tight guarantee was also observed in (Gao et al., 2017) for the simple SBM using different algorithms. We also note that proof techniques like the symmetrization argument (e.g., Hajek et al. (2016)) cannot be directly applied for the sparse regime ($\bar{p} = \mathcal{O}((\log n)/n)$). Recall that we defined the concatenated matrix $\bar{A} = [A_\Gamma^1, A_\Gamma^2, \ldots, A_\Gamma^L]$. Based on the above lemma, we deduce that for any $C > 0$, there exists $C' > 0$ such that

$$||\bar{A} - M_\Gamma|| \leq C'\sqrt{n\bar{p}},$$

with probability at least $1 - \exp(-Cn\bar{p})$.

(b) The second ingredient of the proof is the following lemma, whose proof is provided in Appendix E.6. The lemma provides a lower bound on the distance between two columns of $M_\Gamma$ corresponding to two nodes in distinct clusters.

**Lemma 4.3.** *There exists a constant $C > 0$ such that with probability at least $1 - \exp(-\omega(n))$,*

$$||M_{\Gamma,v} - M_{\Gamma,w}||_2^2 \geq C\varepsilon n\bar{p}^2,$$

*uniformly over all $v, w \in \Gamma$ with $\sigma(v) \neq \sigma(w)$.*

(c) The final proof ingredient, which is proved in Appendix E.7, concerns the performance of the iterative power method with singular value thresholding that outputs the rank-$\hat{K}$ approximation $\hat{A}$ (see Algorithm 2).

**Lemma 4.4.** *For any $c > 0$, there exists a constant $C > 0$ such that with probability at least $1 - 1/n^c$,*

$$||\bar{A} - \hat{A}||_2 \leq C\sigma_{K+1},$$

*where $\sigma_{K+1}$ is the $(K+1)$-th singular value of the matrix $\bar{A}$.*

The challenge is that all these Lemmas require quantified high-probability guarantees. We are now ready to prove the theorem. We first explain why the number of clusters is accurately estimated. It is straightforward to verify that there exist two strictly positive constants, $C_1$ and $C_2$, such that with probability at least $1 - \exp(-\omega(n))$, $C_1\bar{p} \leq \widetilde{p} \leq C_2\bar{p}$ (refer to Lemma C.5). Consequently, from Lemma 4.2, we deduce that for any $C > 0$, with probability at least $1 - \exp(-Cn\bar{p})$, the $(K + 1)$-th singular value of $A_\Gamma$ is significantly smaller than $\sqrt{\widetilde{p}n} \log(n\widetilde{p})$. In conjunction

with Lemma 4.4, this indicates that $K = \hat{K}$ with probability at least $1 - \exp(-Cn\bar{p})$. Therefore, we can assume in the remainder of the proof that $K = \hat{K}$.

Without loss of generality, let us denote $\gamma$ as the permutation of $[K]$ such that the set of misclassified nodes is $\bigcup_{k=1}^K S_k \setminus \mathcal{I}_{\gamma(k)}$. Based on Lemma 4.3, we can prove that: with probability at least $1 - \exp(\omega(n))$,

$$\Big| \bigcup_{k=1}^K S_k \setminus \mathcal{I}_{\gamma(k)} \Big| C'n\bar{p}^2$$

$$\leq \sum_{k=1}^K \sum_{v \in S_k \setminus \mathcal{I}_{\gamma(k)}} ||M_{\Gamma,v} - M_{\Gamma,\gamma(k)}||_2^2$$

$$\leq 8||M_\Gamma - \hat{A}||_F^2 + 8r_{t^*},$$

where $M_{\Gamma,\gamma(k)} = M_{\Gamma,w}$ for $w \in \mathcal{I}_{\gamma(k)}$, and where $r_{t^*}$ is defined in Algorithm 2. Furthermore, for any $C > 0$, using Lemmas 4.2 and 4.4, we can establish that there exists a constant $C_0 > 0$ such that

$$||M_\Gamma - \hat{A}||_F^2 \leq C_0 n\bar{p}$$

with probability at least $1 - \exp(-Cn\bar{p})$. Through a refined analysis of the k-means algorithm, we can also prove the existence of a constant $C_1 > 0$ such that $r_{t^*} \leq C_1 n\bar{p}$. For details, please refer to Appendix D.

### 4.2. Likelihood-Based Improvements

To complete the proof of Theorem 1.2, we analyze the likelihood-based improvement phase of the IAC algorithm. For this purpose, we define the set of well-behaved nodes $H$. We denote the total number of node pairs with observed label $\ell$ including the node $v$ and a node from $S$ by $e(v, S, \ell) = \sum_{w \in S} A_{vw}^\ell$, and let $e(v, S) = \sum_{\ell=1}^L e(v, S, \ell)$. Let $H$ be the largest set of nodes $v \in \mathcal{I}$ that meet the following three conditions with some constant $C_{H1} > 0$:

(H1) $e(v, \mathcal{I}) \leq C_{H1} n\bar{p}$;

(H2) when $v \in \mathcal{I}_k$, $\sum_{i=1}^K \sum_{\ell=0}^L e(v, \mathcal{I}_i, \ell) \log \frac{p(k,i,\ell)}{p(j,i,\ell)} \geq \frac{n\bar{p}}{\log^4 n\bar{p}}$ for all $j \neq k$;

(H3) $e(v, \mathcal{I} \setminus H) \leq \frac{2n\bar{p}}{\log^5(n\bar{p})}$.

We will show that all nodes in $H$ are correctly clustered with high probability, and that the expected number of nodes not in $H$ matches the lower bound on the expected number of misclassified nodes. Each condition in the definition of $H$ can be interpreted as follows: (H1) imposes some regularity in the degree of the node, (H2) implies that $v \in H$ is correctly classified when using the likelihood, and the last condition (H3) implies that the node does not have too many labels pointing outside of the set $H$.

First, we show that the number of nodes not in $H$ can be upper bounded by a number $s$ that is of the same order as $n \exp(-nD(\alpha, p))$, both in expectation and with high probability.

**Proposition 4.5.** *When*

$$s \geq n \exp\left(-nD(\alpha, p) + \frac{n\bar{p}}{\log^3 n\bar{p}}\right), \text{ we have}$$

$$\frac{\mathbb{E}[|\mathcal{I} \setminus H|]}{s} \leq 1 + \exp\left(-\frac{3n\bar{p}}{4\log^3 n\bar{p}}\right) + \exp(-\omega(n\bar{p})).$$

*Moreover,* $\lim_{n\to\infty} \mathbb{P}(|\mathcal{I} \setminus H| \leq s) = 1.$

The proof of Proposition 4.5 can be found in Appendix C.3. The proof reveals that the probability of a node satisfying (H2) is dominant compared to the probabilities of the other two conditions and is of the order of $\exp(-nD(\alpha, p))$.

Next, we examine the performance of the likelihood-based improvement step (Line 6 in the IAC algorithm) for nodes in $H$. In the following proposition, we quantify the improvement achieved with one iteration of this step.

**Proposition 4.6.** *Assume that there exists a constant $C > 0$ such that*

$$\left| \bigcup_{k=1}^{K} (S_k^{(0)} \setminus \mathcal{I}_k) \cap H \right| \leq C \frac{1}{\bar{p}}.$$

*Then, for any constant $C' > 0$, with probability at least $1 - \exp(-C'n\bar{p})$, the following statement holds:*

$$\frac{|\bigcup_{k=1}^{K}(S_k^{(t+1)} \setminus \mathcal{I}_k) \cap H|}{|\bigcup_{k=1}^{K}(S_k^{(t)} \setminus \mathcal{I}_k) \cap H|} \leq \frac{1}{\sqrt{n\bar{p}}} \quad \text{for all} \quad t \geq 0.$$

The proof of Proposition 4.6 can be found in Appendix C.4 and takes advantage of the fact that a likelihood-based test using the estimator $\hat{p}(j, i, \ell)$ matches the test that would use the true likelihood, with high probability. Next, we present the sketch of the proof of Theorem 1.2 (refer to Appendix C.2 for a complete proof).

*Proof sketch of Theorem 1.2.* we can now complete the proof by observing that from Proposition 4.6, after the $\lceil \log n \rceil$ iterations of the likelihood-based improvement step,

$$| \cup_{k=1}^{K} (S_k^{(\lceil \log n \rceil)} \setminus \mathcal{I}_k) \cap H| = 0,$$

with probability at least $1 - \exp(-Cn\bar{p})$ for any constant $C > 0$. Combining this result with Proposition 4.5, when $s \geq n \exp\left(-nD(\alpha, p) + \frac{n\bar{p}}{\log^3 n\bar{p}}\right)$,

$$\mathbb{E}[\varepsilon^{\text{IAC}}(n)] \leq \frac{1}{1 - o(1)} \mathbb{E}[|\mathcal{I} \setminus H|] + o(1)$$

$$\leq \frac{s}{1 - o(1)} \left(1 + \exp\left(-\frac{3n\bar{p}}{4\log^3 n\bar{p}}\right) + \exp(-\omega(n\bar{p}))\right) + o(1)$$

and $\varepsilon^{\text{IAC}}(n) \leq s + o(1)$, with high probability.

### 4.3. Technical Challenges in Instance-Specific Analysis

The design principles behind IAC build upon existing studies that have proposed two-phase algorithms employing initial spectral clustering followed by iterative improvement steps (Yun & Proutiere, 2016; Gao et al., 2017; Gao & Zhang, 2022; Xie & Xu, 2023). However, their analyses are either (i) minimax in nature or limited to the simple SBM—with no analysis for general number $s = o(n)$ (Gao et al., 2017; Gao & Zhang, 2022; Xie & Xu, 2023)—or (ii) provide only high-probability guarantees (Yun & Proutiere, 2016).

Most of the existing analyses are limited to the simple SBM or minimax frameworks and hence do not readily transfer to the instance-specific analysis. If we consider the general (L)SBM, the optimal decision boundary of the cluster assignment is asymmetric and should take into account the information geometry, as reflected in the definition of our divergence $D(\alpha, p)$.

The exception is (Yun & Proutiere, 2016), who provided an instance-optimal analysis under the general LSBM. However, their analysis is limited to high-probability guarantees. More precisely, this algorithm misclassifies fewer than $s$ nodes with a probability that tends to 1 as $n$ grows large, provided that $s$ and the parameters satisfy the condition (1). The probability of the *failure* event (which corresponds to the case where the algorithm misclassifies more than $s$ nodes) is not quantified, but is necessary if one wishes to derive guarantees in expectation.

## 5. Conclusion

In this paper, we investigate the problem of recovering hidden communities in the Labeled Stochastic Block Model (LSBM) with a finite number of clusters whose sizes grow linearly with the total number of nodes. We reveal the necessary and sufficient conditions under which the expected number of misclassified nodes is less than $s$, for any number $s = o(n)$:

$$\liminf_{n\to\infty} \frac{nD(\alpha, p)}{\log(n/s)} \geq 1.$$

To achieve this, we propose IAC (Instance-Adaptive Clustering), the first algorithm whose performance matches these lower bounds both in expectation and with high probability. IAC is a novel two-phase algorithm that consists of a one-shot spectral clustering step followed by iterative likelihood-based cluster assignment improvements. Notably, this approach is based on an instance-specific nonlinear lower bound and does not require any knowledge of the model parameters, including the number of clusters. By performing the spectral clustering only once, IAC maintains an overall computational complexity of $\mathcal{O}(n \operatorname{polylog}(n))$, making it scalable and practical for large-scale problems.

Our results bridge the gap between existing upper and lower bounds for cluster recovery in the LSBM, providing tight necessary and sufficient conditions.

Future work could explore extending these results to models with more complex cluster structures (e.g., the Multiplex Stochastic Block Model) or consider robustness to model misspecification and adversarial noise.

## Acknowledgements

Kaito Ariu is supported by JSPS KAKENHI Grants No. 23K19986 and No. 25K21291. Alexandre Proutiere is supported by the Wallenberg AI, Autonomous Systems and Software Program (WASP) funded by the Knut and Alice Wallenberg Foundation, the Swedish Research Council (VR) and Digital Futures. S. Yun is supported by the Institute of Information & Communications Technology Planning & Evaluation (IITP) grant funded by the Korea government (MSIT) (No. RS-2022-II220311, Development of Goal-Oriented Reinforcement Learning Techniques for Contact-Rich Robotic Manipulation of Everyday Objects, No. RS-2024-00457882, AI Research Hub Project, and No. RS-2019-II190075, Artificial Intelligence Graduate School Program (KAIST)).

## Impact Statement

This paper presents work whose goal is to advance the field of machine learning. There are many potential societal consequences of our work, none of which we feel must be specifically highlighted here.

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

## A. Notation Summary

Please refer to Table 1 for a summary of the notations we use.

*Table 1.* Notations

| Symbol | Description |
|---|---|
| $\mathcal{I}$ | Set of items or nodes |
| $n$ | Number of nodes |
| $K$ | Number of clusters |
| $\mathcal{I}_k$ | Set of nodes in the cluster $k$ |
| $\alpha = (\alpha_1, \alpha_2, \ldots, \alpha_K)$ | Probability that nodes are in each cluster |
| $L$ | Number of labels |
| $\mathcal{L} = \{1, \ldots, L\}$ | Set of labels |
| $p(i, j, \ell)$ | Probability that the label $\ell$ is observed between nodes in $\mathcal{I}_i$ and $\mathcal{I}_j$ |
| $\bar{p}$ | $\max_{i,j,\ell \geq 1} p(i, j, \ell)$ |
| $\eta$ | Positive constant in (A1) |
| $\varepsilon$ | Positive constant in (A2) |
| $\kappa$ | Positive constant in (A3) |
| $D(\alpha, p)$ | Divergence defined as in Introduction |
| $\pi$ | Clustering algorithm |
| $\varepsilon^{\pi}(n)$ | Number of misclassified nodes |
| $I^*(n, a, b)$ | $-2\log\left(\sqrt{\frac{a}{n}}\sqrt{\frac{b}{n}} + \sqrt{1 - \frac{a}{n}}\sqrt{1 - \frac{b}{n}}\right)$ |
| $A^{\ell} = (A_{uv}^{\ell})_{u,v \in \mathcal{I}}$ | Observation matrix for each label $\ell$ |
| $M^{\ell}$ | Expected matrix of $A^{\ell}$ |
| $A_{\Gamma}^{\ell}$ | Trimmed observation matrix, $(A_{\Gamma}^{\ell})_{wv} = A^{\ell}\mathbb{1}_{w,v \in \Gamma}$ |
| $M_{\Gamma}^{\ell}$ | Trimmed expected matrix, $(M_{\Gamma}^{\ell})_{wv} = M^{\ell}\mathbb{1}_{w,v \in \Gamma}$ |
| $\bar{A}$ | $\bar{A} = [A_{\Gamma}^1, A_{\Gamma}^2, \ldots, A_{\Gamma}^L]$ |
| $M, M_{\Gamma}$ | $M = [M^1, M^2, \ldots, M^L], \ M_{\Gamma} = [M_{\Gamma}^1, M_{\Gamma}^2, \ldots, M_{\Gamma}^L]$ |
| $\hat{K}$ | Estimated number of clusters |
| $\hat{A}$ | Rank-$\hat{K}$ approximation of $\bar{A}$ |
| $X_{\Gamma}^{\ell}$ | $A_{\Gamma}^{\ell} - M_{\Gamma}^{\ell}$ |
| $(S_k)_{k \in [\hat{K}]}$ | Initial cluster estimates |
| $(\hat{\mathcal{I}}_k)_{k \in [\hat{K}]}$ | Final cluster estimates |

We use $\|\cdot\|$ to denote the $\ell_2$-norm, i.e., $\|\boldsymbol{x}\| = \sqrt{\sum_i |x_i|}$. We use the standard matrix norm $\|A\| = \sup_{x:\|x\|_2=1} \|Ax\|_2$. We denote by $M^{\ell}$ the expectation of the matrix of $A^{\ell}$, i.e., $M_{u,v}^{\ell} = p(i, j, \ell)$ when $u \in \mathcal{I}_i$ and $v \in \mathcal{I}_j$. Let $M = [M^1, M^2, \ldots, M^L]$. We also denote by $e(v, S, \ell) = \sum_{w \in S} A_{vw}^{\ell}$ the total number of node pairs with observed label $\ell$ including the node $v$ and a node from $S$ and $\mu(v, S, \ell) = \frac{e(v,S,\ell)}{|S|}$ the empirical density of label $\ell$. Let $e(v, S) = \sum_{\ell=1}^{L} e(v, S, \ell)$ and $\mu(v, S) = [\mu(v, S, \ell)]_{0 \leq \ell \leq L}$. In what follows, $e(v, \mathcal{I})$ is referred to as the *degree* of node $v$, which represents the number of observed labels that are different from 0 for pairs of nodes that include $v$.

## B. Lemmas Regarding the Divergence $D(\alpha, p)$

In this part, we present several lemmas in (Yun & Proutiere, 2016) related to the divergence $D(\alpha, p)$.

**Lemma B.1.** *Let $(i^*, j^*)$ be the indices that minimize $D_{L+}(p(i), p(j))$ with $i^* < j^*$. In this case, there exists a $q \in \mathcal{P}^{K \times (L+1)}$ such that*

$$D(\alpha, p) = \sum_{k=1}^{K} \alpha_k \, \mathrm{kl}(q(k), p(i^*, k)) = \sum_{k=1}^{K} \alpha_k \, \mathrm{kl}(q(k), p(j^*, k)).$$

*Proof.* Let us prove the existence of such a $q$ by contradiction. Suppose that

$$D(\alpha, p) = \sum_{k=1}^{K} \alpha_k \, \mathrm{kl}(q(k), p(i^*, k)) > \sum_{k=1}^{K} \alpha_k \, \mathrm{kl}(q(k), p(j^*, k)).$$

In this case, there must be a $k_0$ for which $\mathrm{kl}(q(k_0), p(i^*, k_0)) > \mathrm{kl}(q(k_0), p(j^*, k_0))$. Noting the positive nature of the KL-divergence, $q(k_0) \neq p(i^*, k_0)$. As a result of the KL-divergence's continuity, we can create $q'$ such that $q(k) = q'(k)$ for all $k \neq k_0$, and the following conditions hold: $\mathrm{kl}(q(k_0), p(i^*, k_0)) - \epsilon < \mathrm{kl}(q'(k_0), p(i^*, k_0)) < \mathrm{kl}(q(k_0), p(i^*, k_0))$ and $\mathrm{kl}(q'(k_0), p(j^*, k_0)) < \mathrm{kl}(q(k_0), p(j^*, k_0)) + \epsilon$ for some $0 < \epsilon < (\mathrm{kl}(q(k_0), p(i^*, k_0)) - \mathrm{kl}(q(k_0), p(j^*, k_0)))/2$. With this selection of $q'$, we obtain:

$$D(\alpha, p) > \sum_{k=1}^{K} \alpha_k \, \mathrm{kl}(q'(k), p(i^*, k)) > \sum_{k=1}^{K} \alpha_k \, \mathrm{kl}(q'(k), p(j^*, k)),$$

which is in contradiction with the definition of $D(\alpha, p)$. ∎

**Lemma B.2.** *When $\bar{p} = o(1)$,*

$$\lim_{n \to \infty} \frac{D(\alpha, p)}{\sum_{k=1}^{K} \frac{\alpha_k}{2} \left( \sum_{\ell=1}^{L} (\sqrt{p(i^*, k, \ell)} - \sqrt{p(j^*, k, \ell)})^2 \right)} \geq 1.$$

*Proof.* Let $(i^*, j^*)$ be the pair that minimizes $D_{L+}(\alpha, p(i), p(j))$ with $i^* < j^*$. According to Lemma B.1, there exists $q$ such that

$$D(\alpha, p) = \sum_{k=1}^{K} \alpha_k \, \mathrm{kl}(q(k), p(i^*, k)) = \sum_{k=1}^{K} \alpha_k \, \mathrm{kl}(q(k), p(j^*, k)).$$

Following this,

$$
\begin{aligned}
nD(\alpha, p) &= n \frac{\sum_{k=1}^{K} (\alpha_k \, \mathrm{kl}(q(k), p(i^*, k)) + \alpha_k \, \mathrm{kl}(q(k), p(j^*, k)))}{2} \\
&= -n \sum_{k=1}^{K} \alpha_k \sum_{\ell=0}^{L} q(k, \ell) \log \left( \frac{\sqrt{p(i^*, k, \ell) p(j^*, k, \ell)}}{q(k, \ell)} \right) \\
&\geq n \sum_{k=1}^{K} \alpha_k \sum_{\ell=0}^{L} \left( q(k, \ell) - \sqrt{p(i^*, k, \ell) p(j^*, k, \ell)} \right) \\
&= n \sum_{k=1}^{K} \alpha_k \left( \frac{\sum_{\ell=1}^{L} (p(i^*, k, \ell) + p(j^*, k, \ell))}{2} - \sum_{\ell=1}^{L} \sqrt{p(i^*, k, \ell) p(j^*, k, \ell)} \right) (1 - o(1)) \\
&= n \sum_{k=1}^{K} \frac{\alpha_k}{2} \left( \sum_{\ell=1}^{L} (\sqrt{p(i^*, k, \ell)} - \sqrt{p(j^*, k, \ell)})^2 \right) (1 - o(1)).
\end{aligned}
$$

∎

**Lemma B.3.** *Under (A1), when $\bar{p} = o(1)$, $\limsup_{n \to \infty} \frac{D(\alpha, p)}{\eta \bar{p} L} \leq 1$.*

*Proof.* Based on the definition of $D(\alpha, p)$, for any $i \neq j$, we have:

$$
\begin{aligned}
D(\alpha, p) &\leq \max \left\{ \sum_{k=1}^{K} \alpha_k \, \mathrm{kl}(p(i, k), p(i, k)), \sum_{k=1}^{K} \alpha_k \, \mathrm{kl}(p(i, k), p(j, k)) \right\} \\
&= \sum_{k=1}^{K} \alpha_k \, \mathrm{kl}(p(i, k), p(j, k)) \\
&\leq \sum_{k=1}^{K} \alpha_k \sum_{\ell=1}^{L} \frac{(p(i, k, \ell) - p(j, k, \ell))^2}{p(j, k, \ell)} (1 + o(1))
\end{aligned}
$$

$$\leq \quad \sum_{k=1}^{K} \alpha_k \sum_{\ell=1}^{L} \eta \bar{p} (1 + o(1))$$
$$= \quad \eta \bar{p} L (1 + o(1)),$$

where we employ $\log(1 + x) = x(1 + o(1))$ when $x = o(1)$. ∎

## C. Performance Analysis of the IAC Algorithm – Proof of Theorem 1.2

### C.1. Preliminaries

**Lemma C.1** (Chernoff-Hoeffding theorem)**.** *Let* $X_1, \ldots, X_n$ *be i.i.d. Bernoulli random variables with mean* $\nu$*. Then, for any* $\delta > 0$,

$$\mathbb{P}\left(\frac{1}{n} \sum_{i=1}^{n} X_i \geq \nu + \delta\right) \leq \exp\left(-n \, \mathrm{kl}(\nu + \delta, \nu)\right)$$

$$\mathbb{P}\left(\frac{1}{n} \sum_{i=1}^{n} X_i \leq \nu - \delta\right) \leq \exp\left(-n \, \mathrm{kl}(\nu - \delta, \nu)\right)$$

**Lemma C.2** (Pinsker's inequality (Tsybakov, 2008))**.** *For any* $0 \leq p, q \leq 1$, $2(p - q)^2 \leq \mathrm{kl}(p, q)$.

**Lemma C.3.** *For each* $v \in \mathcal{I}$*, for any constant* $C > 0$,

$$\mathbb{P}\left(e(v, \mathcal{I}) \geq C n \bar{p}\right) \geq \exp(-(C - e^L) n \bar{p}).$$

The proof is given in Appendix E.1.

**Lemma C.4.** *For all* $v \in \mathcal{I}_k$ *and* $D \geq 0$,

$$\mathbb{P}\left\{ \left(\sum_{i=1}^{K} |\mathcal{I}_i| \, \mathrm{kl}(\mu(v, \mathcal{I}_i), p(k, i)) \geq nD\right) \cap \left(e(v, \mathcal{I}) \leq C \eta n \bar{p}\right) \right\}$$
$$\leq \exp\left(-nD + KL \log(10 \eta L n \bar{p}) + \frac{100 \eta^2 n \bar{p}^2 L^2}{\alpha_1}\right)$$

The proof is given in Appendix E.2.

Regarding the estimation of $\bar{p}$ as $\widetilde{p}$, we show that $\widetilde{p}$ has the same order as $\bar{p}$ with high probability:

**Lemma C.5.** *Let* $C_1$ *and* $C_2$ *be constants such that* $0 < C_1 < \frac{1}{\eta}\left(1 - \frac{1}{e^L}\right)$ *and* $C_2 > e^L$*. Then,*

$$C_1 \bar{p} \leq \widetilde{p} \leq C_2 \bar{p}$$

*holds with probability at least* $1 - \exp\left(-\omega(n)\right)$.

The proof is given in Appendix E.3.

### C.2. Proof of Theorem 1.2

For each $k \in [K]$, from Chernoff-Hoeffding's theorem and Pinsker's inequality, for any constant $C > 0$, we get:

$$\mathbb{P}\left(||\mathcal{I}_k| - \alpha_k n| \leq \sqrt{n} \log n|\right) \leq \exp\left(-n \, \mathrm{kl}\left(\alpha_k - \frac{\log n}{\sqrt{n}}, \alpha_k\right)\right) + \exp\left(-n \, \mathrm{kl}\left(\alpha_k + \frac{\log n}{\sqrt{n}}, \alpha_k\right)\right)$$
$$\leq 2 \exp\left(-2(\log n)^2\right)$$
$$\leq \frac{1}{n^C}. \tag{2}$$

Hence, we make the assumption that for every $k \in [K]$, the inequality

$$||\mathcal{I}_k| - \alpha_k n| \leq \sqrt{n} \log n \tag{3}$$

is maintained throughout the remainder of the proof.

With a positive constant $C_{H1}$, let $H$ be the largest set of nodes $v \in \mathcal{I}$ satisfying:

(H1)  $e(v, \mathcal{I}) \leq C_{\text{H1}} n\bar{p}$.

(H2)  When $v \in \mathcal{I}_k$, $\sum_{i=1}^{K} \sum_{\ell=0}^{L} e(v, \mathcal{I}_i, \ell) \log \frac{p(k,i,\ell)}{p(j,i,\ell)} \geq \frac{n\bar{p}}{\log(n\bar{p})^4}$ for all $j \neq k$.

(H3)  $e(v, \mathcal{I} \setminus H) \leq \frac{2n\bar{p}}{\log^5(n\bar{p})}$.

(H1) controls degrees, (H2) implies that $v \in H$ is accurately classified using the (true) log-likelihood, and (H3) indicates that $v$ has a limited number of shared labels with nodes not in $H$.

From Proposition 4.6 and Theorem 4.1, after the $\lceil \log n \rceil$ iterations in the further improvement step (remember that $n\bar{p} = \omega(1)$, so that $1/\sqrt{n\bar{p}} \leq e^{-2}$ when $n$ is large enough), for any $c > 0$, there exists $C > 0$ such that

$$
\begin{aligned}
|\bigcup_{k=1}^{K} (S_k^{(\lceil \log n \rceil)} \setminus \mathcal{I}_k) \cap H| &\leq \frac{1}{(n\bar{p})^{\frac{\log n}{2}}} |\bigcup_{k=1}^{K} (S_k^{(0)} \setminus \mathcal{I}_k) \cap H| \\
&\leq \frac{1}{e^{\log n}} \cdot \frac{C}{\bar{p}} \\
&= o(1)
\end{aligned}
\tag{4}
$$

with probability at least $1 - \exp(-cn\bar{p})$, where the last equality is from $n\bar{p} = \omega(1)$. Therefore, for any $C > 0$, no node in $H$ can be misclassified with probability at least $1 - \exp(-Cn\bar{p})$. Hence the number of misclassified nodes cannot exceed $|\mathcal{I} \setminus H|$, when $s \geq n \exp\left(-nD(\alpha, p) + \frac{n\bar{p}}{\log^3 n\bar{p}}\right)$ (the condition of Proposition 4.5 holds). When $s \geq n \exp\left(-nD(\alpha, p) + \frac{n\bar{p}}{\log^3 n\bar{p}}\right)$,

$$1 \leq \liminf_{n \to \infty} \frac{nD(\alpha, p) - \frac{n\bar{p}}{\log(n\bar{p})^3}}{\log(n/s)} = \liminf_{n \to \infty} \frac{nD(\alpha, p)}{\log(n/s)}, \tag{5}$$

where we used $D(\alpha, p) = \Omega(\bar{p})$ (from (A2) and Lemma B.2).

Let $\mathcal{A}$ be an event defined as:

$$\mathcal{A} = \left\{ |\cup_{k=1}^{K} (S_k^{(0)} \setminus \mathcal{I}_k) \cap H| \leq C\frac{1}{\bar{p}} \text{ and (3) hold} \right\},$$

where $C > 0$ is some large enough constant. We upper bound the expected number of misclassified nodes as follows.

$$
\begin{aligned}
\mathbb{E}[\varepsilon^{\text{IAC}}(n)] &= \mathbb{E}[\varepsilon^{\text{IAC}}(n)|\mathcal{A}]\mathbb{P}(\mathcal{A}) + \mathbb{E}[\varepsilon^{\text{IAC}}(n)|\bar{\mathcal{A}}]\mathbb{P}(\bar{\mathcal{A}}) \\
&\leq \mathbb{E}[\varepsilon^{\text{IAC}}(n)|\mathcal{A}]\mathbb{P}(\mathcal{A}) + n\mathbb{P}(\bar{\mathcal{A}}) \\
&\overset{(a)}{\leq} \mathbb{E}[\varepsilon^{\text{IAC}}(n)|\mathcal{A}] + o(1) \\
&\overset{(b)}{\leq} \mathbb{E}\left[|\mathcal{I} \setminus H| \big| \mathcal{A}\right] + o(1) \\
&= \sum_{x=0,\ldots,n} x\mathbb{P}(|\mathcal{I} \setminus H| = x|\mathcal{A}) + o(1) \\
&= \sum_{x=0,\ldots,n} x\frac{\mathbb{P}(|\mathcal{I} \setminus H| = x, \mathcal{A})}{\mathbb{P}(\mathcal{A})} + o(1)
\end{aligned}
$$

$$\leq \frac{1}{\mathbb{P}(\mathcal{A})} \sum_{x=0,\dots,n} x \mathbb{P}(|\mathcal{I} \setminus H| = x) + o(1)$$

$$\overset{(c)}{\leq} \frac{1}{1 - o(1)} \mathbb{E}[|\mathcal{I} \setminus H|] + o(1)$$

$$\overset{(d)}{\leq} \frac{1}{1 - o(1)} s \left( 1 + \exp\left( -\frac{3n\bar{p}}{4 \log^3 n\bar{p}} \right) + \exp(-\omega(n\bar{p})) \right) + o(1),$$

where $(a)$ is from Theorem 4.1, (2), and $\bar{p} = O(\log n / n)$; $(b)$ is from Proposition 4.6; $(c)$ is from Theorem 4.1; and $(d)$ is from Proposition 4.5. This concludes the proof.

∎

### C.3. Proof of Proposition 4.5

We quantify the number of nodes satisfying (H1) and (H2) in (6) and (7), respectively.

Number of nodes satisfying (H1): From Lemma C.3, for any constant $C_{\text{H1}} > 0$, for each $v \in \mathcal{I}$,

$$\mathbb{P}\{e(v, \mathcal{I}) \leq C_{\text{H1}} n\bar{p}\} \geq 1 - \exp(-(C_{\text{H1}} - e^L) n\bar{p}). \tag{6}$$

Number of nodes satisfying (H2): We aim to prove that when $v$ fulfills (H1), it also satisfies (H2) with a probability of at least

$$1 - \exp\left( -nD(\alpha, p) + \frac{n\bar{p}}{2 \log(n\bar{p})^3} \right) - \exp(-\omega(n\bar{p})). \tag{7}$$

To achieve this, we first assert that if $v$ meets the condition

$$\sum_{i=1}^{K} |\mathcal{I}_i| \, \text{kl}(\mu(v, \mathcal{I}_i), p(k, i)) \leq \left( 1 - \frac{\log(n)^2}{\sqrt{n}} \right) nD(\alpha, p) - \frac{n\bar{p}}{\log(n\bar{p})^4}, \tag{8}$$

then $v$ complies with (H2). In fact, assuming that (8) is true, we have the following:

(i) $\sum_{i=1}^{K} \alpha_i n \, \text{kl}(\mu(v, \mathcal{I}_i), p(k, i)) \leq \left( 1 + \frac{\log(n)^2}{\sqrt{n}} \right) \sum_{i=1}^{K} |\mathcal{I}_i| \, \text{kl}(\mu(v, \mathcal{I}_i), p(k, i)) < nD(\alpha, p)$, because $||\mathcal{I}_i| - \alpha_i n| \leq \sqrt{n} \log(n)$ (from (2)) and (8) holds with probability at least $1 - \exp(-\omega(n\bar{p}))$;

(ii) $\sum_{i=1}^{K} \alpha_i n \, \text{kl}(\mu(v, \mathcal{I}_i), p(j, i)) \geq nD(\alpha, p)$, since
$\max\left\{ \sum_{i=1}^{K} \alpha_i \, \text{kl}(\mu(v, \mathcal{I}_i), p(j, i)), \sum_{i=1}^{K} \alpha_i \, \text{kl}(\mu(v, \mathcal{I}_i), p(k, i)) \right\} \geq D(\alpha, p)$ and
$\sum_{i=1}^{K} \alpha_i \, \text{kl}(\mu(v, \mathcal{I}_i), p(k, i)) < D(\alpha, p)$;

(iii) $\sum_{i=1}^{K} |\mathcal{I}_i| \, \text{kl}(\mu(v, \mathcal{I}_i), p(j, i)) \geq \left( 1 - \frac{\log(n)^2}{\sqrt{n}} \right) nD(\alpha, p)$, from (ii) and the fact that $||\mathcal{I}_i| - \alpha_i n| \leq \sqrt{n} \log(n)$;

(iv) from (8) and (iii), for all $j \neq i$,

$$\begin{aligned} \sum_{i=1}^{K} \sum_{\ell=0}^{L} e(v, \mathcal{I}_i, \ell) \log \frac{p(k, i, \ell)}{p(j, i, \ell)} &= \sum_{i=1}^{K} |\mathcal{I}_i| \left( \text{kl}(\mu(v, \mathcal{I}_i), p(j, i)) - \text{kl}(\mu(v, \mathcal{I}_i), p(k, i)) \right) \\ &\geq \frac{n\bar{p}}{\log(n\bar{p})^4}. \end{aligned}$$

Therefore, $v$ satisfies (H2). The remaining task is to assess the probability of event (8), which can be done by applying Lemma C.4 and proving (7).

Based on (6) and (7), the expected number of nodes that fail to meet either (H1) or (H2) can be upper bounded as follows.

$$\mathbb{E}[\text{The number of nodes that do not satisfy either (H1) or (H2)}]$$

$$\leq n\exp(-(C_{\mathrm{H1}} - e^L)n\bar{p}) + n\exp\left(-nD(\alpha, p) + \frac{n\bar{p}}{2\log^3 n\bar{p}}\right) + n\exp(-\omega(n\bar{p})).$$

From Markov's inequality,

$$\mathbb{P}\left(\text{The number of nodes that do not satisfy either (H1) or (H2)} \geq \frac{1}{\bar{p}(n\bar{p})^5}\right)$$

$$\leq (n\bar{p})^6 \exp(-(C_{\mathrm{H1}} - e^L)n\bar{p}) + (n\bar{p})^6 \exp\left(-nD(\alpha, p) + \frac{n\bar{p}}{2\log^3 n\bar{p}}\right) + (n\bar{p})^6 \exp(-\omega(n\bar{p}))$$

$$\leq \exp\left(-nD(\alpha, p) + \frac{n\bar{p}}{4\log^3 n\bar{p}}\right).$$

We obtain the following upper bound on the expected number of $|\mathcal{I} \setminus H|$.

$$\frac{\mathbb{E}[|\mathcal{I} \setminus H|]}{s} \leq 1 + \frac{n\exp\left(-nD(\alpha, p) + \frac{n\bar{p}}{4\log(n\bar{p})^3}\right) + n\exp\left(-\omega(n\bar{p})\right)}{s}$$

$$\leq 1 + \exp\left(-\frac{3n\bar{p}}{4\log^3 n\bar{p}}\right) + \exp(-\omega(n\bar{p})).$$

This concludes the proof of the guarantee in expectation as $n\bar{p} = \omega(1)$.

Regarding the high probability guarantee, the subsequent Lemma C.6 is instrumental in finalizing the proof, and its proof can be found in Appendix E.4.

**Lemma C.6.** *Let* $\phi \leq 1/(\bar{p}(n\bar{p})^5)$. *When the number of nodes that do not satisfy either (H1) or (H2) is less than* $\phi/3$, $|\mathcal{I} \setminus H| \leq \phi$, *with probability at least* $1 - \exp\left(-\omega(n\bar{p})\right)$.

From Markov's inequality, for a sufficiently large choice of $C_{\mathrm{H1}}$,

$$\mathbb{P}\left(\text{The number of nodes that do not satisfy either (H1) or (H2)} \geq s/3\right)$$

$$\leq \frac{\mathbb{E}[\text{The number of nodes that do not satisfy either (H1) or (H2)}]}{s/3}$$

$$\leq \frac{n\exp(-(C_{\mathrm{H1}} - e^L)n\bar{p}) + n\exp\left(-nD(\alpha, p) + \frac{n\bar{p}}{2\log^3 n\bar{p}}\right) + n\exp(-\omega(n\bar{p}))}{s/3}$$

$$\leq \frac{n\exp(-(C_{\mathrm{H1}} - e^L)n\bar{p}) + n\exp\left(-nD(\alpha, p) + \frac{n\bar{p}}{2\log^3 n\bar{p}}\right) + n\exp(-\omega(n\bar{p}))}{\frac{n}{3}\exp\left(-nD(\alpha, p) + \frac{n\bar{p}}{\log^3 n\bar{p}}\right)}$$

$$\leq 6\exp(-(C_{\mathrm{H1}} - e^L)n\bar{p} + nD(\alpha, p)) + 3\exp\left(-\frac{n\bar{p}}{2\log^3 n\bar{p}}\right) + 3\exp(-\omega(n\bar{p}))$$

$$\overset{(a)}{=} o(1),$$

where for $(a)$, we used $nD(\alpha, p) = \mathcal{O}(n\bar{p})$ from Lemma B.3. Combining with Lemma C.6, $|\mathcal{I} \setminus H| \leq s$ with high probability. This concludes the proof.

∎

## C.4. Proof of Proposition 4.6

Recall that $\{S_j^{(t)}\}_{1 \leq j \leq K}$ represents the partition after the $t$-th improvement iteration. Note that without loss of generality, we assume the set of misclassified nodes in $H$ after the $t$-th step to be $\mathcal{E}^{(t)} = \left(\cup_k (S_k^{(t)} \setminus \mathcal{I}_k)\right) \cap H$ (it should be defined through an appropriate permutation $\gamma$ of $\{1, \ldots, K\}$ as $\mathcal{E}^{(t)} = (\cup_k (S_k^{(t)} \setminus \mathcal{I}_{\gamma(k)})) \cap H$, but we omit $\gamma$). With this notation,

we can define $\mathcal{E}_{jk}^{(t)} = (S_j^{(t)} \cap \mathcal{I}_k) \cap H$ and $\mathcal{E}^{(t)} = \bigcup_{j,k:j\neq k} \mathcal{E}_{jk}^{(t)}$. During each improvement step, nodes move to the most likely cluster (according to the log-likelihood defined in the IAC algorithm). As a result, for all $i \in [K]$,

$$0 \leq \sum_{j,k:j\neq k} \sum_{v\in\mathcal{E}_{jk}^{(t+1)}} \sum_{i=1}^{K}\sum_{\ell=0}^{L} e(v,S_i^{(t)},\ell) \log \frac{\hat{p}(j,i,\ell)}{\hat{p}(k,i,\ell)}$$

$$\leq \sum_{j,k:j\neq k} \sum_{v\in\mathcal{E}_{jk}^{(t+1)}} \sum_{i=1}^{K}\sum_{\ell=0}^{L} e(v,S_i^{(t)},\ell) \log \frac{p(j,i,\ell)}{p(k,i,\ell)} + |\mathcal{E}^{(t+1)}|(n\bar{p})^{1-\kappa}\log(n\bar{p})^3 \tag{9}$$

$$\leq \sum_{j,k:j\neq k} \sum_{v\in\mathcal{E}_{jk}^{(t+1)}} \sum_{i=1}^{K}\sum_{\ell=0}^{L} e(v,\mathcal{I}_i,\ell) \log \frac{p(j,i,\ell)}{p(k,i,\ell)}$$

$$+ \sum_{w\in\mathcal{E}^{(t+1)}} e(w,\mathcal{E}^{(t)})\log(2\eta) + 5|\mathcal{E}^{(t+1)}|\frac{n\bar{p}}{\log^5 n\bar{p}} \tag{10}$$

$$\leq -\frac{n\bar{p}}{\log^4 n\bar{p}}|\mathcal{E}^{(t+1)}| + \sum_{w\in\mathcal{E}^{(t+1)}} e(w,\mathcal{E}^{(t)},\ell)\log(2\eta) + 5|\mathcal{E}^{(t+1)}|\frac{n\bar{p}}{\log^5 n\bar{p}} \tag{11}$$

$$\leq -\frac{n\bar{p}}{\log^4 n\bar{p}}|\mathcal{E}^{(t+1)}| + \sqrt{|\mathcal{E}^{(t)}||\mathcal{E}^{(t+1)}|n\bar{p}\log n\bar{p}} + 6|\mathcal{E}^{(t+1)}|\frac{n\bar{p}}{\log^5 n\bar{p}}. \tag{12}$$

Hence, based on the aforementioned inequalities, we deduce that

$$\frac{|\mathcal{E}^{(t+1)}|}{|\mathcal{E}^{(t)}|} \leq \frac{\log^{11} n\bar{p}}{n\bar{p}} \leq \frac{1}{\sqrt{n\bar{p}}}.$$

Subsequently, we will validate each step involved in the prior analysis.

*Proof of* (9)*:* Utilizing the inequality $\log(1+x) \leq x$, when $p(j,i,\ell) - |\hat{p}(j,i,\ell) - p(j,i,\ell)| > 0$,

$$\left|\log\frac{\hat{p}(j,i,\ell)}{p(j,i,\ell)}\right| \leq \frac{|\hat{p}(j,i,\ell) - p(j,i,\ell)|}{p(j,i,\ell) - |\hat{p}(j,i,\ell) - p(j,i,\ell)|}.$$

Therefore, to prove (9), we merely need to present an upper bound for $|\hat{p}(j,i,\ell) - p(j,i,\ell)|$. Applying the triangle inequality,

$$|\hat{p}(j,i,\ell) - p(j,i,\ell)|$$
$$= \frac{\left|e(S_i^{(0)},S_j^{(0)},\ell) - p(j,i,\ell)|S_i^{(0)}||S_j^{(0)}|\right|}{|S_i^{(0)}||S_j^{(0)}|}$$
$$\leq \frac{\left|e(S_i^{(0)},S_j^{(0)},\ell) - \mathbb{E}[e(S_i^{(0)},S_j^{(0)},\ell)]\right| + \left|\mathbb{E}[e(S_i^{(0)},S_j^{(0)},\ell)] - p(j,i,\ell)|S_i^{(0)}||S_j^{(0)}|\right|}{|S_i^{(0)}||S_j^{(0)}|}. \tag{13}$$

Firstly, we determine an upper bound for $\left|e(S_i^{(0)},S_j^{(0)},\ell) - \mathbb{E}[e(S_i^{(0)},S_j^{(0)},\ell)]\right|$. Let $\mathcal{S}$ represent a partition such that

$$\left|\bigcup_{k=1}^{K}\mathcal{I}_k \setminus S_k\right| \leq \xi = O\left(\frac{\log(n\bar{p})^2}{\bar{p}}\right) \quad \text{for all} \quad \{S_k\}_{1\leq k\leq K} \in \mathcal{S}.$$

Following this,

$$|\mathcal{S}| \leq \binom{n}{\xi}K^{\xi}$$
$$\leq \left(\frac{Ken}{\xi}\right)^{\xi}$$

$$= \exp\left(O\left(\frac{\log(n\bar{p})^3}{\bar{p}}\right)\right). \tag{14}$$

For every $\{S_k\}_{1\leq k\leq K} \in \mathcal{S}$ and for all $\ell \geq 1$ and $1 \leq i,j \leq K$, $e(S_i, S_j, \ell)$ represents the sum of $|S_i||S_j|$ (or $\frac{|S_i|^2}{2}$ when $i = j$) independent Bernoulli random variables. Given that the variance of $e(S_i, S_j, \ell)$ is less than $n^2\bar{p}$, by applying the Chernoff inequality (for example, Theorem 2.1.3 in (Tao, 2012)), with a probability of at least $1 - \exp\left(-\Theta\left(\frac{\log(n\bar{p})^4}{\bar{p}}\right)\right)$,

$$|e(S_i, S_j, \ell) - \mathbb{E}[e(S_i, S_j, \ell)]| \leq n\log(n\bar{p})^2 \quad \text{for all} \quad i,j,\ell. \tag{15}$$

From (14) and (15) and union bound, with probability at least $1 - \exp\left(-\Theta\left(\frac{\log(n\bar{p})^4}{\bar{p}}\right)\right)$,

$$|e(S_i, S_j, \ell) - \mathbb{E}[e(S_i, S_j, \ell)]| \leq n\log(n\bar{p})^2 \quad \text{for all} \quad i,j,\ell \quad \text{and} \quad \{S_k\}_{1\leq k\leq K} \in \mathcal{S}.$$

Since $\{S_k^{(0)}\}_{1\leq k\leq K} \in \mathcal{S}$, from the above inequality,

$$\left|e(S_i^{(0)}, S_j^{(0)}, \ell) - \mathbb{E}[e(S_i^{(0)}, S_j^{(0)}, \ell)]\right| \leq n\log(n\bar{p})^2 \quad \text{for all} \quad i,j,\ell, \tag{16}$$

with at least probability $1 - \exp\left(-\Theta\left(\frac{\log(n\bar{p})^4}{\bar{p}}\right)\right)$.

We now devote to the remaining part of (13). Since for any $C > 0$, $|\mathcal{E}^{(0)}| = O\left(\frac{\log(n\bar{p})^2}{\bar{p}}\right)$ with at least probability $1 - \exp\left(-Cn\bar{p}\right)$ from Theorem 4.1,

$$\left|\mathbb{E}[e(S_i^{(0)}, S_j^{(0)}, \ell)] - |S_i^{(0)}||S_j^{(0)}|p(i,j,\ell)\right| \leq \eta|\mathcal{E}^{(0)}|np(i,j,\ell) = O(n\log(n\bar{p})^2). \tag{17}$$

Observe that since $\bar{p} = \mathcal{O}(\frac{\log n}{n})$ and $n\bar{p} = \omega(1)$,

$$\frac{n\bar{p}}{\left(\frac{\log(n\bar{p})^4}{\bar{p}}\right)} = \mathcal{O}\left(\frac{(\log n)n\bar{p}}{n\log(n\bar{p})^4}\right)$$

$$= \mathcal{O}\left(\frac{\log n^2}{n\log(\log n)^4}\right)$$

$$= o(1). \tag{18}$$

Considering (13), (16), (17), and (18), for any $C > 0$, with a probability of at least $1 - \exp\left(-Cn\bar{p}\right)$,

$$|\hat{p}(j,i,\ell) - p(j,i,\ell)| = O(\log(n\bar{p})^2/n) \quad \text{for all} \quad i,j,\ell,$$

which leads to the conclusion that:

$$\left|\log\frac{\hat{p}(j,i,\ell)}{p(j,i,\ell)}\right| \leq \frac{|\hat{p}(j,i,\ell) - p(j,i,\ell)|}{p(j,i,\ell) - |\hat{p}(j,i,\ell) - p(j,i,\ell)|} = O\left(\frac{\log(n\bar{p})^2}{np(j,i,\ell)}\right) \quad \text{for all} \quad i,j,\ell.$$

Given that $e(v, S_i^{(t)}, \ell) \leq e(v,\mathcal{I}) \leq 10\eta n\bar{p}L$ as per (H1) and $np(j,i,\ell) \geq (n\bar{p})^\kappa$ according to (A3), for all $v \in \Gamma$ and $i,j,k$,

$$\sum_{\ell=0}^{L} e(v, S_i^{(t)}, \ell)\left|\log\frac{\hat{p}(j,i,\ell)}{\hat{p}(k,i,\ell)} - \log\frac{p(j,i,\ell)}{p(k,i,\ell)}\right| = O\left(\log(n\bar{p})^2(n\bar{p})^{1-\kappa}\right).$$

*Proof of (10):* As $\log\frac{p(j,i,0)}{p(k,i,0)} = O(\bar{p})$ holds for all $i,j,k$ and $|\mathcal{E}^{(t)}| = O(\log(n\bar{p})^2/\bar{p})$,

$$\sum_{i=1}^{K}\sum_{\ell=0}^{L} e(v, S_i^{(t)}, \ell)\log\frac{p(j,i,\ell)}{p(k,i,\ell)}$$

$$= \sum_{i=1}^{K} \left( |S_i^{(t)}| \log \frac{p(j,i,0)}{p(k,i,0)} + \sum_{\ell=1}^{L} e(v, S_i^{(t)}, \ell) \log \frac{p(j,i,\ell)p(k,i,0)}{p(k,i,\ell)p(j,i,0)} \right)$$

$$\leq \sum_{i=1}^{K} \left( |\mathcal{I}_i| \log \frac{p(j,i,0)}{p(k,i,0)} + \sum_{\ell=1}^{L} e(v, S_i^{(t)}, \ell) \log \frac{p(j,i,\ell)p(k,i,0)}{p(k,i,\ell)p(j,i,0)} \right) + \log(n\bar{p})^3$$

$$\leq \sum_{i=1}^{K} \sum_{\ell=0}^{L} e(v, \mathcal{I}_i, \ell) \log \frac{p(j,i,\ell)}{p(k,i,\ell)} + \sum_{i=1}^{K} \sum_{\ell=1}^{L} e(v, \mathcal{I}_i \setminus S_i^{(t)}, \ell) \log(2\eta) + \log(n\bar{p})^3$$

$$= \sum_{i=1}^{K} \sum_{\ell=0}^{L} e(v, \mathcal{I}_i, \ell) \log \frac{p(j,i,\ell)}{p(k,i,\ell)} + \left( e(v, \mathcal{E}^{(t)}) + e(v, \mathcal{I} \setminus H) \right) \log(2\eta) + \log(n\bar{p})^3$$

$$\leq \sum_{i=1}^{K} \sum_{\ell=0}^{L} e(v, \mathcal{I}_i, \ell) \log \frac{p(j,i,\ell)}{p(k,i,\ell)} + \log(2\eta)e(v, \mathcal{E}^{(t)}) + \frac{4n\bar{p}}{\log^5 n\bar{p}},$$

where the last inequality arises from (H3), specifically, from the condition $e(v, \mathcal{I} \setminus H) \leq \frac{2n\bar{p}}{\log^5 n\bar{p}}$ when $v$ belongs to $H$.

*Proof of* (11)*:* Given that $\mathcal{E}^{(t+1)} \subset H$ and all $v \in H$ fulfill (H2), every $v$ in $\mathcal{E}_{jk}^{(i+1)}$ meets the following condition:

$$\sum_{i=1}^{K} \sum_{\ell=0}^{L} e(v, \mathcal{I}_i, \ell) \log \frac{p(j,i,\ell)}{p(k,i,\ell)} \leq -\frac{n\bar{p}}{\log^4 n\bar{p}}.$$

*Proof of* (12)*:* Define $\bar{\Gamma} = \{v : e(v, \mathcal{I}) \leq C_{\text{H1}} n\bar{p}\}$ and let $A_{\bar{\Gamma}}^{\ell}$ represent the modified matrix of $A^{\ell}$ with elements in rows and columns corresponding to $w \notin \bar{\Gamma}$ set to 0. $\bar{\Gamma}$ consists of all nodes that meet (H1), and $H$ is a subset of $\bar{\Gamma}$. Consider $X_{\bar{\Gamma}} = \sum_{\ell=1}^{L}(A_{\bar{\Gamma}}^{\ell} - M_{\bar{\Gamma}}^{\ell})$. We obtain:

$$\sum_{v \in \mathcal{E}^{(t+1)}} \left( e(v, \mathcal{E}^{(t)}) - \mathbb{E}[e(v, \mathcal{E}^{(t)})] \right) \leq 1_{\mathcal{E}^{(t)}}^{T} \cdot X_{\bar{\Gamma}} \cdot 1_{\mathcal{E}^{(t+1)}},$$

where $1_S$ denotes a vector where the $v$-th component is 1 if $v \in S$ and 0 otherwise. Given that $\mathbb{E}[e(v, \mathcal{E}^{(t)})] \leq \bar{p}L|\mathcal{E}^{(t)}|$ and for any $C > 0$, the $\|X_{\bar{\Gamma}}\|_2 \leq \sqrt{n\bar{p}\log n\bar{p}}$ holds with a probability of at least $1 - \exp(-Cn\bar{p})$ according to Lemma 4.2,

$$\sum_{v \in \mathcal{E}^{(t+1)}} e(v, \mathcal{E}^{(t)}) = \sum_{v \in \mathcal{E}^{(t+1)}} \left( e(v, \mathcal{E}^{(t)}) - \mathbb{E}[e(v, \mathcal{E}^{(t)})] \right) + \bar{p}L|\mathcal{E}^{(t)}||\mathcal{E}^{(t+1)}|$$

$$\leq \|1_{\mathcal{E}^{(t)}}^{T} \cdot X_{\bar{\Gamma}} \cdot 1_{\mathcal{E}^{(t+1)}}\|_2 + |\mathcal{E}^{(t+1)}|\log(n\bar{p})$$

$$\leq \|1_{\mathcal{E}^{(t)}}^{T}\|_2 \|X_{\bar{\Gamma}}\|_2 \|1_{\mathcal{E}^{(t+1)}}\|_2 + |\mathcal{E}^{(t+1)}|\log(n\bar{p})$$

$$\leq \sqrt{|\mathcal{E}^{(t)}||\mathcal{E}^{(t+1)}|n\bar{p}\log(n\bar{p})} + |\mathcal{E}^{(t+1)}|\log(n\bar{p}).$$

This concludes the proof. $\blacksquare$

## D. Proof of Theorem 4.1

First, we show a spectral analysis on $A_{\bar{\Gamma}}^{\ell} - M_{\bar{\Gamma}}^{\ell}$ by extending the technique by (Feige & Ofek, 2005; Coja-Oghlan, 2010).

**Lemma D.1.** *For any $\ell \in [L]$, for any $C > 0$, there exists $C' > 0$ such that:*

$$\|X_{\bar{\Gamma}}^{\ell}\|_2 \leq C'\sqrt{n\bar{p}},$$

*with probability at least $1 - \exp(-Cn\bar{p})$.*

The proof of Lemma 4.2 is given in Appendix E.5.

Therefore, for any $C > 0$, there exists $C' > 0$ such that:

$$\|\bar{A} - M_\Gamma\| \leq \sum_{\ell=1}^{L} \|X_\Gamma^\ell\|$$
$$\leq C'\sqrt{n\bar{p}},$$

with probability at least $1 - \exp(-Cn\bar{p})$. Next, we prove the lower bound on the column distance of $M_\Gamma$.

**Lemma D.2.** *There exists a constant $C > 0$ such that with probability at least $1 - \exp(-\omega(n))$,*

$$\|M_{\Gamma,v} - M_{\Gamma,w}\|_2^2 \geq C\varepsilon n\bar{p}^2,$$

*uniformly over all $v, w \in \Gamma$ with $\sigma(v) \neq \sigma(w)$.*

The proof of Lemma 4.3 is given in Appendix E.6.

Furthermore, we can show that the iterative power method with the singular value thresholding procedure estimates the number of clusters correctly and the matrix $\hat{A}$ is an accurate rank-$K$ approximation of the matrix $\bar{A}$ with sufficiently high probability.

**Lemma D.3.** *For any $c > 0$, there exists a constant $C > 0$ such that with probability at least $1 - 1/n^c$,*

$$\|\bar{A} - \hat{A}\|_2 \leq C\sigma_{K+1},$$

*where $\sigma_{K+1}$ is the $(K + 1)$-th singular value of the matrix $\bar{A}$.*

The proof is in Appendix E.7.

We can prove the theorem as follows. The proof draws inspiration from the proof of Theorem 4 in (Yun & Proutiere, 2014b). However, we obtain a high probability certificate of $1 - \exp(-Cn\bar{p})$ for any $C > 0$. Throughout this section, we assume $\hat{K} = K$ (for any $c > 0$, the event occurs with probability at least $1 - \exp(-cn\bar{p})$ from Theorem 4.1) and let $\gamma(k) := \arg\min_\theta \left| \bigcup_{k=1}^{K} S_k \setminus \mathcal{I}_{\theta(k)} \right|$. Define $M_{\Gamma,\gamma(k)}$ as $M_{\Gamma,\gamma(k)} := M_{\Gamma,v}$, where $v$ is a node that satisfies $v \in \mathcal{I}_{\gamma(k)}$. According to Lemma 4.3, there exists a constant $C' > 0$ such that, with probability at least $1 - \exp(-\omega(n))$,

$$\varepsilon^\pi(n) \cdot C'n\bar{p}^2 = \left| \bigcup_{k=1}^{K} S_k \setminus \mathcal{I}_{\gamma(k)} \right| \cdot C'n\bar{p}^2$$

$$\leq \sum_{k=1}^{K} \sum_{v \in S_k \setminus \mathcal{I}_{\gamma(k)}} \|M_{\Gamma,v} - M_{\Gamma,\gamma(k)}\|_2^2$$

$$\leq 2\sum_{k=1}^{K} \sum_{v \in S_k \setminus \mathcal{I}_{\gamma(k)}} \left( \|M_{\Gamma,v} - \xi_k\|_2^2 + \|\xi_k - M_{\Gamma,\gamma(k)}\|_2^2 \right)$$

$$\leq 4\sum_{k=1}^{K} \sum_{v \in S_k \setminus \mathcal{I}_{\gamma(k)}} \|M_{\Gamma,v} - \xi_k\|_2^2$$

$$\leq 8\sum_{k=1}^{K} \sum_{v \in S_k \setminus \mathcal{I}_{\gamma(k)}} \left( \|M_{\Gamma,v} - \hat{A}_v\|_2^2 + \|\hat{A}_v - \xi_k\|_2^2 \right)$$

$$\leq 8\|M_\Gamma - \hat{A}\|_F^2 + 8r_{t^*},$$

where $\|\cdot\|_F$ denotes the Frobenius norm of a matrix. To complete the proof, we need:

$$\|M_\Gamma - \hat{A}\|_F^2 = \mathcal{O}(n\bar{p}) \tag{19}$$
$$r_{t^*} = \mathcal{O}(n\bar{p}), \tag{20}$$

with the high probability guarantee.

We first prove (19). From the equation (1.4) of (Halko et al., 2011),

$$\sigma_{k+1} = \min_{\mathrm{rank}(X) \leq k} \|A - X\|_2,$$

where $\sigma_k$ is the $k$-th largest singular value of the matrix $A$.

For any matrix $A$ of rank $K$, it holds that $\|A\|_F^2 \leq K\|A\|_2^2$. Since the rank of the matrix $\hat{A}$ and $M_\Gamma$ are $K$, the rank of the matrix $\hat{A} - M_\Gamma$ is at most $2K$. Then, Lemma 4.4 implies, for any $c > 0$, there exist constants $C_1, C_2 > 0$ such that with probability at least $1 - 1/n^c$,

$$
\begin{aligned}
\|M_\Gamma - \hat{A}\|_F^2 &\leq 2K\|M_\Gamma - \hat{A}\|_2^2 \\
&\leq 4K\|M_\Gamma - A_\Gamma\|_2^2 + 4K\|A_\Gamma - \hat{A}\|_2^2 \\
&\leq 4K\|M_\Gamma - A_\Gamma\|_2^2 + 4KC_1\sigma_{K+1}^2, \\
&= 4K\|M_\Gamma - A_\Gamma\|_2^2 + 4KC_1\left(\min_{\mathrm{rank}(X) \leq K}\|A_\Gamma - X\|_2\right)^2 \\
&\leq 4K\|M_\Gamma - A_\Gamma\|_2^2, + 4KC_1\|M_\Gamma - A_\Gamma\|_2^2 \\
&\leq C_2 K\|M_\Gamma - A_\Gamma\|_2^2.
\end{aligned}
$$

Therefore, together with Lemma 4.2, for any $c > 0$, there exists a constant $C > 0$ such that

$$\|M_\Gamma - \hat{A}\|_F^2 \leq Cn\bar{p}$$

with probability at least $1 - \exp(-cn\bar{p})$.

Next, we prove (20). It is sufficient to show that there exists $i_t \in \{1, 2, ..., \lfloor \log n \rfloor\}$ such that $r_t = \mathcal{O}(n\bar{p})$. First, by Lemma 4.2, for any $C > 0$, there exists a positive constant $D_1 > 0$ such that

$$\|\hat{A} - M_\Gamma\|_2^2 \leq 8D_1 n\bar{p} \tag{21}$$

with probability at least $1 - \exp(-Cn\bar{p})$. By Lemma C.5, for any constant $D_2 > \frac{20K}{\alpha_1}$, there exists $i_t \in \{1, ..., \lfloor \log n \rfloor\}$ such that with probability at least $1 - \exp(-\omega(n))$,

$$32D_2 D_1 \bar{p} \leq i_t \frac{\widetilde{p}}{100} \leq \frac{64\eta}{1 - 1/e} D_2 D_1 \bar{p}$$

Define sets of nodes $I_k$ and $W$ as

$$
\begin{aligned}
I_k &= \left\{ v \in \mathcal{I}_k \cap \Gamma : \|\hat{A}_v - M_{\Gamma,k}\|_2^2 \leq \frac{1}{4}i_t \frac{\widetilde{p}}{100} \right\} \\
W &= \left\{ v \in \Gamma : \|\hat{A}_v - M_{\Gamma,k}\|_2^2 \geq 4i_t \frac{\widetilde{p}}{100}, \text{ for all } k \in [K] \right\}.
\end{aligned}
$$

These sets have the following properties:

- For all $v \in I_k$, $I_k \subset Q_v^{(i_t)}$
- For all $v, v' \in I_k$, $\|\hat{A}_v - \hat{A}_{v'}\|_2^2 \leq 2\|\hat{A}_v - M_{\Gamma,k}\|_2^2 + 2\|\hat{A}_{v'} - M_{\Gamma,k}\|_2^2 \leq i_t \frac{\widetilde{p}}{100}$
- For all $v \in W$, $\left(\bigcup_{k=1}^K I_k\right) \cap Q_v^{(i_t)} = \emptyset$
  - since for all $v' \in \cup_{k=1}^K I_k$, $\|\hat{A}_v - \hat{A}_{v'}\|_2^2 \geq \frac{1}{2}\|\hat{A}_v - M_{\Gamma,\sigma(v')}\|_2^2 - \|\hat{A}_{v'} - M_{\Gamma,\sigma(v')}\|_2^2 > i_t \frac{\widetilde{p}}{100}$.

From (21), we have

$$\left| \Gamma \setminus \left( \bigcup_{k=1}^K I_k \right) \right| \left( \frac{1}{4}i_t \frac{\widetilde{p}}{100} \right) \leq \sum_{v \in \Gamma \setminus \left( \cup_{k=1}^K I_k \right)} \|\hat{A}_v - M_{\Gamma,\sigma(v)}\|_2^2$$

$$\leq \|\hat{A} - M_\Gamma\|_F^2$$
$$\leq 2K\|\hat{A} - M_\Gamma\|_2^2$$
$$\leq 16KD_1 n\bar{p},$$

with probability at least $1 - \exp(-Cn\bar{p})$. Thus, we have

$$\left| \Gamma \setminus \left( \bigcup_{k=1}^K I_k \right) \right| \leq 16KD_1 n\bar{p} \left( \frac{1}{4} i_t \frac{\widetilde{p}}{100} \right)^{-1}$$
$$\leq \frac{2nK}{D_2}$$
$$< n\frac{\alpha_1}{10},$$

with probablity at least $1 - \exp(-Cn\bar{p}(1 + o(1)))$. Therefore, we can deduce that

- For all $v \in W$, $|Q_v^{(i_t)}| \leq n\frac{\alpha_1}{10}$

- For all $v \in \bigcup_{k=1}^K I_k$, $|Q_v^{(i_t)}| \geq \alpha_1 n - \lfloor n\exp(-n\widetilde{p}) \rfloor - n\frac{\alpha_1}{10} \geq \frac{4}{5}\alpha_1 n$

with probability at least $1 - \exp(-Cn\bar{p}(1 + o(1)))$.

For each $k \in [K]$, the probability that $\mathcal{V}_R \cap I_k = \emptyset$ is at most:

$$\left( 1 - \frac{4}{5}\alpha_1 \right)^{\lceil (\log n)^2 \rceil} \leq \frac{1}{n^{C\log n}},$$

where $C > 0$ is a constant depends only on $\alpha_1$. Thus, for any $c' > 0$, $\mathcal{V}_R$ contains at least one node from $I_k$ for all $k \in [K]$ with probability at least $1 - \exp(-c'n\bar{p}(1 + o(1)))$. Therefore, any $v \in W$ will not be assigned as $v_k^*$, with probability at least $1 - \exp(-Cn\bar{p}(1 + o(1)))$. It implies that $\|\hat{A}_{v_k^*} - M_{\Gamma,\gamma(k)}\|_2^2 \leq 4i_t\frac{\widetilde{p}}{100}$ for all $k \in [K]$ with probability at least $1 - \exp(-Cn\bar{p}(1 + o(1)))$. Regarding the centroid $\xi_k^{i_t}$ of the clusters, since $\hat{A}_v$ is within $\sqrt{5i_t\frac{\widetilde{p}}{100}}$ from $M_{\Gamma,\gamma(k)}$ in the Euclidean distance for all $v \in T_k^{i_t}$,

$$\|\xi_k^{i_t} - M_{\Gamma,\gamma(k)}\|_2^2 \leq 5i_t\frac{\widetilde{p}}{100}$$
$$= \frac{320\eta D_1 D_2}{1 - 1/e}\bar{p}, \quad \forall k \in [K],$$

with probability at least $1 - \exp(-Cn\bar{p}(1 + o(1)))$. Thus,

$$r_{i_t} = \sum_{k=1}^K \sum_{v \in T_k^{(i_t)}} \|\hat{A}_v - \xi_k^{(i_t)}\|_2^2$$
$$\leq \sum_{k=1}^K \sum_{v \in \mathcal{I}_k \cap \Gamma} \|\hat{A}_v - \xi_k^{(i_t)}\|_2^2$$
$$\leq 2\sum_{k=1}^K \sum_{v \in \mathcal{I}_k \cap \Gamma} \left( \|\hat{A}_v - M_{\Gamma,v}\|_2^2 + \|\xi_k^{(i_t)} - M_{\Gamma,v}\|_2^2 \right)$$
$$\leq \left( 32KD_1 + \frac{640\eta D_1 D_2}{1 - 1/e} \right) n\bar{p},$$

with probability at least $1 - \exp(-Cn\bar{p}(1 + o(1)))$. This concludes the proof of Theorem 4.1. ∎

# E. Proofs of Lemmas and Corollary

## E.1. Proof of Lemma C.3

Let $\{X_i\}$ be Bernoulli i.i.d. random variable with mean $\bar{p}$. First, for any $C_1 > 0$ and for every $v \in \mathcal{I}$, by Markov's inequality,

$$
\begin{aligned}
\mathbb{P}\left\{e(v,\mathcal{I}) \geq C_1 n\bar{p}\right\} &\leq \inf_{\lambda \geq 0} \frac{\mathbb{E}[\exp(\lambda e(v,\mathcal{I}))]}{\exp(\lambda C_1 n\bar{p})} \\
&\leq \inf_{\lambda \geq 0} \frac{\mathbb{E}\left[\exp(\lambda L \sum_{i=1}^{n} X_i)\right]}{\exp(\lambda C_1 n\bar{p})} \\
&= \inf_{\lambda \geq 0} \frac{\prod_{i=1}^{n}(\bar{p}(\exp(\lambda L)-1)+1)}{\exp(\lambda C_1 n\bar{p})} \\
&\leq \inf_{\lambda \geq 0} \frac{\prod_{i=1}^{n}(\bar{p}\exp(\lambda L)+1)}{\exp(\lambda C_1 n\bar{p})} \\
&\leq \inf_{\lambda \geq 0} \frac{\prod_{i=1}^{n}\exp(\bar{p}\exp(\lambda L))}{\exp(\lambda C_1 n\bar{p})} \\
&\leq \inf_{\lambda \geq 0} \frac{\exp(n\bar{p}\exp(\lambda L))}{\exp(\lambda C_1 n\bar{p})} \\
&\leq \exp(-(C_1 - e^L)n\bar{p})
\end{aligned}
$$

This concludes the proof. ∎

## E.2. Proof of Lemma C.4

Consider $\mathcal{X}$ as a collection of $K \times (L+1)$ matrices, defined as follows:

$$
\mathcal{X} = \left\{ \boldsymbol{x} \in \mathbb{Z}^{K \times (L+1)} : \quad \sum_{i=1}^{K}\sum_{\ell=1}^{L} x_{i,\ell} \leq 10\eta n\bar{p}L, \quad \text{and} \quad \sum_{\ell=0}^{L} x_{i,\ell} = |\mathcal{I}_i| \quad \text{for all} \quad 1 \leq i \leq K \right\}.
$$

To simplify notation, we employ $[\frac{x_{i,\ell}}{|\mathcal{I}_i|}]$ in place of $[\frac{x_{i,\ell}}{|\mathcal{I}_i|}]_{0 \leq \ell \leq L}$ to denote the probability mass vector for labels defined by $x_i$. We also use $e(v)$ to represent the $K \times (L+1)$ matrix, where the $(i,\ell)$-th element corresponds to $e(v,\mathcal{I}_i,\ell)$. Consequently, for $v \in \mathcal{I}_k$,

$$
\begin{aligned}
&\mathbb{P}\left\{ \left( \sum_{i=1}^{K} |\mathcal{I}_i|\,\mathrm{kl}(\mu(v,\mathcal{I}_i), p(k,i)) \geq nD \right) \cap \left( e(v,\mathcal{I}) \leq 10n\bar{p}L \right) \right\} \\
&= \sum_{\boldsymbol{x} \in \mathcal{X}} \mathbb{P}\left\{e(v) = \boldsymbol{x}\right\} \mathbb{P}\left\{ \sum_{i=1}^{K} |\mathcal{I}_i|\,\mathrm{kl}(\mu(v,\mathcal{I}_i), p(k,i)) \geq nD \,\middle|\, e(v) = \boldsymbol{x} \right\} \\
&\leq \sum_{\boldsymbol{x} \in \mathcal{X}} \mathbb{P}\{e(v) = \boldsymbol{x}\} \frac{\exp\left( \sum_{i=1}^{K} |\mathcal{I}_i|\,\mathrm{kl}([\frac{x_{i,\ell}}{|\mathcal{I}_i|}], p(k,i)) \right)}{\exp(nD)} \\
&\leq \sum_{\boldsymbol{x} \in \mathcal{X}} \mathbb{P}\{e(v) = \boldsymbol{x}\} \frac{\prod_{i=1}^{K}\prod_{\ell=0}^{L}\left( \frac{x_{i,\ell}}{|\mathcal{I}_i|p(k,i,\ell)} \right)^{x_{i,\ell}}}{\exp(nD)} \\
&\overset{(a)}{\leq} \frac{1}{\exp(nD)} \sum_{\boldsymbol{x} \in \mathcal{X}} \prod_{i=1}^{K}\left( \left(1 - \frac{\sum_{\ell=1}^{L} x_{i,\ell}}{|\mathcal{I}_i|}\right)^{x_{i,0}} \exp\left(\sum_{\ell=1}^{L} x_{i,\ell}\right) \right) \\
&= \frac{1}{\exp(nD)} \sum_{\boldsymbol{x} \in \mathcal{X}} \prod_{i=1}^{K} \exp\left( (|\mathcal{I}_i| - \sum_{\ell=1}^{L} x_{i,\ell})\log\left(1 - \frac{\sum_{\ell=1}^{L} x_{i,\ell}}{|\mathcal{I}_i|}\right) + \sum_{\ell=1}^{L} x_{i,\ell} \right) \\
&\leq \frac{1}{\exp(nD)} \sum_{\boldsymbol{x} \in \mathcal{X}} \prod_{i=1}^{K} \exp\left( \frac{(\sum_{\ell=1}^{L} x_{k,\ell})^2}{|\mathcal{I}_i|} \right)
\end{aligned}
$$

$$\leq \frac{(10\eta n\bar{p}L)^{KL} \exp(100\eta^2 n\bar{p}^2 L^2/\alpha_1)}{\exp(nD)}$$

$$= \exp\left(-nD + KL\log(10\eta Ln\bar{p}) + \frac{100\eta^2 n\bar{p}^2 L^2}{\alpha_1}\right),$$

where $(a)$ comes from the subsequent inequality:

$$\mathbb{P}\{e(v, \mathcal{I}_i, \ell) = x_{i,\ell} \quad \text{for all} \quad i, \ell\} \leq \prod_{i=1}^{K}\left(p(k, i, 0)^{x_{i,0}} \prod_{\ell=1}^{L} \binom{|\mathcal{I}_i|}{x_{i,\ell}} p(k, i, \ell)^{x_{k,\ell}}\right)$$

$$\leq \prod_{i=1}^{K}\left(p(k, i, 0)^{x_{i,0}} \prod_{\ell=1}^{L}\left(\frac{e|\mathcal{I}_i|}{x_{i,\ell}}\right)^{x_{i,\ell}} p(k, i, \ell)^{x_{i,\ell}}\right).$$

This concludes the proof. ∎

### E.3. Proof of Lemma C.5

First, we evaluate the probability of the event $\widetilde{p} \leq C_2\bar{p}$ does not hold. Let $\{X_i\}$ be Bernoulli i.i.d. random variables with mean $\bar{p}$. We have:

$$\mathbb{P}\{\widetilde{p} \geq C_2\bar{p}\} = \mathbb{P}\left\{\frac{2}{n(n-1)}\sum_{\ell=1}^{L}\sum_{v,w\in\mathcal{I}:v>w} A_{vw}^{\ell} \geq C_2\bar{p}\right\}$$

$$\leq \mathbb{P}\left\{\frac{2L}{n(n-1)}\sum_{i=1}^{n(n-1)/2} X_i \geq C_2\bar{p}\right\}$$

$$\leq \inf_{\lambda\geq 0} \frac{\mathbb{E}\exp\left(\lambda L\sum_{i=1}^{n(n-1)/2} X_i\right)}{\exp\left(\lambda\frac{n(n-1)}{2}C_2\bar{p}\right)}$$

$$= \inf_{\lambda\geq 0} \frac{\prod_{i=1}^{n(n-1)/2}\left(p(\exp(\lambda L)-1)+1\right)}{\exp\left(\lambda\frac{n(n-1)}{2}C_2\bar{p}\right)}$$

$$\leq \inf_{\lambda\geq 0} \frac{\prod_{i=1}^{n(n-1)/2}\left(p\exp(\lambda L)+1\right)}{\exp\left(\lambda\frac{n(n-1)}{2}C_2\bar{p}\right)}$$

$$\leq \inf_{\lambda\geq 0} \frac{\exp\left(\frac{n(n-1)}{2}p\exp(\lambda L)\right)}{\exp\left(\lambda\frac{n(n-1)}{2}C_2\bar{p}\right)}$$

$$\leq \exp\left(-(C_2-e^L)\frac{n(n-1)}{2}\bar{p}\right)$$

$$\leq \exp\left(-\frac{C_2-e^L}{4}n^2\bar{p}\right). \tag{22}$$

Next, we evaluate the probability that the event $C_1\bar{p} \leq \widetilde{p}$ does not hold. Let $\underline{p} := \min_{i,j,\ell\geq 1} p(i, j, \ell)$. Let $\{Y_i\}$ be Bernoulli i.i.d. random variables with mean $\underline{p}$. We have:

$$\mathbb{P}\{\widetilde{p} \leq C_1\bar{p}\} = \mathbb{P}\left\{-\frac{2}{n(n-1)}\sum_{\ell=1}^{L}\sum_{v,w\in\mathcal{I}:v>w} A_{vw} \geq -C_1\bar{p}\right\}$$

$$\leq \mathbb{P}\left\{-\frac{2}{n(n-1)}L\sum_{i=1}^{n(n-1)/2} Y_i \geq -C_1\bar{p}\right\}$$

$$\leq \inf_{\lambda \geq 0} \frac{\mathbb{E} \exp\left(-\lambda L \sum_{i=1}^{n(n-1)/2} Y_i\right)}{\exp\left(-\lambda \frac{n(n-1)}{2} C_1 \bar{p}\right)}$$

$$\leq \inf_{\lambda \geq 0} \frac{\prod_{i=1}^{n(n-1)/2} \left(\underline{p}\left(\exp(-\lambda L) - 1\right) + 1\right)}{\exp\left(-\lambda \frac{n(n-1)}{2} C_1 \bar{p}\right)}$$

$$\leq \inf_{\lambda \geq 0} \frac{\exp\left(\frac{n(n-1)}{2} \underline{p}\left(\exp(-\lambda L) - 1\right)\right)}{\exp\left(-\lambda \frac{n(n-1)}{2} C_1 \bar{p}\right)}$$

$$\leq \exp\left(-\left(1 - \frac{1}{e^L}\right) \frac{n(n-1)}{2} \underline{p} + \frac{n(n-1)}{2} C_1 \bar{p}\right)$$

$$\leq \exp\left(-\left(1 - \frac{1}{e^L}\right) \frac{n(n-1)}{2} \frac{1}{\eta} \bar{p} + \frac{n(n-1)}{2} C_1 \bar{p}\right)$$

$$= \exp\left(-\left(\left(1 - \frac{1}{e^L}\right) \frac{1}{\eta} - C_1\right) \frac{n(n-1)}{2} \bar{p}\right)$$

$$\leq \exp\left(-\left(\left(1 - \frac{1}{e^L}\right) \frac{1}{\eta} - C_1\right) \frac{n^2}{4} \bar{p}\right) . \tag{23}$$

Combining (22) and (23), we conclude the proof. ∎

### E.4. Proof of Lemma C.6

Define $e(S, S) = \sum_{v \in S} e(S, S)$. We now aim to prove the following intermediate assertion: with high probability, no subset $S \subset \mathcal{I}$ exists such that $e(S, S) \geq \phi \frac{n\bar{p}}{\log^5 n\bar{p}}$ and $|S| = \phi$. For any subset $S \subset \mathcal{I}$ with $|S| = \phi$, using Markov's inequality,

$$\mathbb{P}\left\{e(S, S) \geq \phi \frac{n\bar{p}}{\log^5 n\bar{p}}\right\} \leq \inf_{t \geq 0} \frac{\mathbb{E}[\exp(e(S, S)\phi)]}{\phi t \frac{n\bar{p}}{\log^5 n\bar{p}}}$$

$$\leq \inf_{t \geq 0} \frac{\prod_{i=1}^{\phi^2/2}(1 + L\bar{p}\exp(t))}{\phi t \frac{n\bar{p}}{\log^5 n\bar{p}}}$$

$$\leq \inf_{t \geq 0} \exp\left(\frac{\phi^2 L\bar{p}}{2}\exp(t) - \phi t \frac{n\bar{p}}{\log^5 n\bar{p}}\right)$$

$$\overset{(a)}{\leq} \exp\left(-\frac{\phi n\bar{p}}{\log^5 n\bar{p}}\left(\log\left(\frac{2n}{\phi L \log^5 n\bar{p}}\right) - 1\right)\right)$$

$$\overset{(b)}{\leq} \exp\left(-\frac{\phi n\bar{p}}{2\log^5 n\bar{p}}\log\left(\frac{2n}{\phi L \log^5 n\bar{p}}\right)\right), \tag{24}$$

where we set $t = \log(2n/(\phi L \log^5 n\bar{p}))$ for inequality $(a)$ and utilize $\log\left(2n/(\phi L \log^5 n\bar{p})\right) = \omega(1)$ derived from $\phi \leq 1/(\bar{p}(n\bar{p})^5)$ for inequality $(b)$. Considering the number of subsets $S \subset \mathcal{I}$ having size $\phi$ is $\binom{n}{\phi} \leq (\frac{en}{\phi})^\phi$, we can infer the following from (24):

$$\mathbb{E}\left[\left|\left\{S : e(S, S) \geq \frac{\phi n\bar{p}}{(\log n\bar{p})^5} \text{ and } |S| = s\right\}\right|\right] \leq \left(\frac{en}{\phi}\right)^\phi \exp\left(-\frac{\phi n\bar{p}}{2\log^5 n\bar{p}}\log\left(\frac{2n}{\phi L \log^5 n\bar{p}}\right)\right)$$

$$\leq \exp\left(\phi \log\left(\frac{en}{\phi}\right) - \frac{\phi n\bar{p}}{2\log^5 n\bar{p}}\log\left(\frac{2n}{\phi L \log^5 n\bar{p}}\right)\right)$$

$$\leq \exp\left(-\frac{\phi n\bar{p}}{4\log^5 n\bar{p}}\log\left(\frac{2n}{\phi L \log^5 n\bar{p}}\right)\right) .$$

Hence, applying the Markov inequality, there are no subsets $S \subset \mathcal{I}$ such that $e(S, S) \geq \phi n\bar{p}/(\log n\bar{p})^5$ and $S = \phi$ with a probability of at least $1 - \exp\left(-\frac{\phi n\bar{p}}{4\log^5 n\bar{p}}\log\left(\frac{2n}{\phi L \log^5 n\bar{p}}\right)\right)$.

In order to complete the proof of the lemma, we construct the following series of sets. Let $Z_1$ represent the set of nodes that do not fulfill at least one of (H1) and (H2). Generate the sequence $\{Z(t) \subset \mathcal{I}\}_{1 \leq t \leq t^*}$ as follows:

- $Z(0) = Z_1$.
- For $t \geq 1$, $Z(t) = Z(t-1) \cup \{v_t\}$ if $v_t \in \mathcal{I}$ exists such that $e(v_t, Z(t-1)) > \frac{2n\bar{p}}{\log^5(n\bar{p})}$ and $v_t \notin Z(t-1)$. If no such node exists, the sequence terminates.

The sequence concludes after constructing $Z(t^*)$, which, according to the definition of (H3), is equivalent to $\mathcal{I} \setminus H$. We now demonstrate that if we assume the number of nodes that do not satisfy (H3) is strictly greater than $\phi/2$, then one of the sets in the sequence $\{Z(t) \subset \mathcal{I}\}_{1 \leq t \leq t^*}$ contradicts the claim we just established.

Suppose the number of nodes that do not meet (H3) is strictly greater than $\phi/2$. At some point, these nodes will be incorporated into the sets $Z(t)$, and according to the definition, each of these nodes contributes over $\frac{2n\bar{p}}{\log^5(n\bar{p})}$ to $e(Z(t), Z(t))$. Therefore, if we start with $Z_1$ and add $\phi/2$ nodes that do not satisfy (H3), we obtain a set $Z(t)$ with a cardinality smaller than $\phi/3 + \phi/2$ and such that $e(Z(t), Z(t)) > \frac{2n\bar{p}}{\log^5(n\bar{p})}$. We can further include arbitrary nodes to $Z(t)$ so that its cardinality becomes $\phi$, resulting in a set that contradicts the claim. ∎

### E.5. Proof of Lemma 4.2

We remark that the statement is taken directly from (Le et al., 2017, Theorem 2.1), but here we present an alternative proof which may be of independent interest. We will extend the proof strategy of (Feige & Ofek, 2005). Let us define a discretized space $T$ that approximates the continuous unit sphere. Let $\varepsilon \in (0, 1)$ be a fixed constant.

$$T := \left\{ x \in \left( \frac{\varepsilon}{\sqrt{n}} \mathbb{Z} \right)^n : \sum_i^n x_i = 0, \|x\|_2 \leq 1 \right\},$$

where $\mathbb{Z}$ is the set of integers. From Claim 2.9 in (Feige & Ofek, 2005), $|T| \leq \exp(C(\varepsilon)n)$ where $C(\varepsilon)$ is a constant only depends on $\varepsilon$.

We first aim to prove that for all $x, y \in T$, for any constant $C > 0$, there exists a constant $C' > 0$ such that

$$|x^\top (A_\Gamma^\ell - M_\Gamma^\ell) y| = C' \sqrt{n\bar{p}},$$

with probability at least $1 - \exp(-Cn\bar{p})$. Using this result and Claim 2.4 in (Feige & Ofek, 2005), we can deduce that for any $C > 0$, there exists a constant $C' > 0$ such that

$$\|A_\Gamma^\ell - M_\Gamma^\ell\|_2 = \sup_{\|x\|_2 = 1, \|y\|_2 = 1} |x^\top (A_\Gamma^\ell - M_\Gamma^\ell) y|$$

$$\leq \frac{1}{(1-\varepsilon)^2} C' \sqrt{n\bar{p}},$$

with probability at least $1 - \exp(-Cn\bar{p})$.

Define a set of light couples as:

$$\mathcal{L} = \left\{ (v, w) \in \mathcal{I} \times \mathcal{I} : |x_v y_w| \leq \sqrt{\frac{\bar{p}}{n}} \right\}.$$

Also define a set of heavy couples as its complement:

$$\mathcal{L}^c = \left\{ (v, w) \in \mathcal{I} \times \mathcal{I} : |x_v y_w| > \sqrt{\frac{\bar{p}}{n}} \right\}.$$

Finally, let we define a subset of edges $\mathcal{K} := (\Gamma^c \times \mathcal{I}) \cup (\mathcal{I} \times \Gamma^c)$[2]. Using these sets, by the triangular inequality,

$$|x^\top (A_\Gamma^\ell - M_\Gamma^\ell) y|$$

---

[2]Note that this set is the complement of $\Gamma \times \Gamma$

$$= |x^\top (A_\Gamma^\ell - M^\ell) y + x^\top (M^\ell - M_\Gamma^\ell) y|$$

$$= | \sum_{(v,w) \in \mathcal{L}} x_v A_{vw}^\ell y_w - \sum_{(v,w) \in \mathcal{K} \cap \mathcal{L}} x_v A_{vw}^\ell y_w + \sum_{(v,w) \in \mathcal{L}^c} x_v A_{\Gamma,vw}^\ell y_w - x^\top M^\ell y + x^\top (M^\ell - M_\Gamma^\ell) y|$$

$$\leq P_1(x,y) + P_2(x,y) + P_3(x,y) + P_4(x,y) ,$$

where

$$P_1(x,y) = \left| \sum_{(v,w) \in \mathcal{K} \cap \mathcal{L}} x_v A_{vw}^\ell y_w \right|$$

$$P_2(x,y) = \left| \sum_{(v,w) \in \mathcal{L}} x_v A_{vw}^\ell y_w - x^\top M^\ell y \right|$$

$$P_3(x,y) = \left| \sum_{(v,w) \in \mathcal{L}^c} x_v A_{\Gamma,vw}^\ell y_w \right|$$

$$P_4(x,y) = |x^\top (M^\ell - M_\Gamma^\ell) y| .$$

We will show that for each fixed $x, y \in T$, for any constant $C_1 > 0$,

$$P_1(x,y) \leq 2C_1 \sqrt{n\bar{p}},$$

with probability at least $1 - \exp(-C_1 n)$. We will also show that for each fixed $x, y \in T$, for any $C_2 > 3$,

$$P_2(x,y) \leq C_2 \sqrt{n\bar{p}}$$

with probability at least $1 - \exp\left(-\frac{C_2 - 3}{2} n\right)$. Therefore, as $|T| \leq \exp(C(\varepsilon)n)$, taking the union bound over $T$ yields,

$$P_1(x,y) \leq 2C_1 \sqrt{n\bar{p}} \quad \text{and}$$
$$P_2(x,y) \leq C_2 \sqrt{n\bar{p}},$$

with probability at least $1 - \exp(-(C_1 - C(\varepsilon))n) - \exp\left(-\frac{C_2 - 3 - 2C(\varepsilon)}{2} n\right)$. Therefore, for any constant $C > 0$, there exists constants $C_1$ and $C_2$ such that for all $x, y \in T$,

$$P_1(x,y) \leq C_1 \sqrt{n\bar{p}}$$
$$P_2(x,y) \leq C_2 \sqrt{n\bar{p}},$$

with probability at least $1 - \exp(-Cn)$. Moreover, by extending the argument in (Feige & Ofek, 2005), we will prove that for any constant $C > 0$, there exists a constant $C_3 > 0$ such that

$$P_3(x,y) \leq C_3 \sqrt{n\bar{p}} ,$$

with probability at least $1 - \exp(-Cn\bar{p})$. Furthermore, we will prove that for any constant $C_4 > e$, for all $x, y \in T$,

$$P_4(x,y) \leq n\bar{p} \exp(-C_4 n\bar{p}) ,$$

with probability at least $1 - \exp(-\omega(n))$. Summarizing the bounds altogether using the union bound, we get the statement of the lemma.

From now on, we will focus on proving each bound with the probability guarantees.

**Bound on $P_1(x,y)$:** By Lemma C.5, with constant $c > e$, we have $\lfloor n \exp(-n\widetilde{p}) \rfloor \leq n \exp(-cn\bar{p})$ with probability at least $1 - \exp(-\omega(n))$. Let $\{X_i\}$ be i.i.d. Bernoulli random variables with mean $\bar{p}$. When, $\lfloor n \exp(-n\widetilde{p}) \rfloor \leq n \exp(-cn\bar{p})$, the following inequalities hold:

$$\mathbb{P} \left\{ \sum_{(v,w) \in \Gamma^c \times \mathcal{I}} A_{vw}^\ell \geq C_1 n \ \middle| \ \lfloor n \exp(-n\widetilde{p}) \rfloor \leq n \exp(-cn\bar{p}) \right\}$$

$$= \mathbb{P}\left\{ 2 \sum_{(v,w)\in\Gamma^c\times\mathcal{I}: v>w} A_{vw}^\ell \geq C_1 n \;\middle|\; \lfloor n\exp(-n\widetilde{p})\rfloor \leq n\exp(-cn\bar{p}) \right\}$$

$$\leq \mathbb{P}\left\{ \sum_{i=1}^{n^2\exp(-cn\bar{p})} X_i \geq C_1\frac{n}{2} \right\}$$

$$\leq \inf_{\lambda\geq 0} \frac{\mathbb{E}\left[\exp(\lambda\sum_{i=1}^{n^2\exp(-cn\bar{p})} X_i)\right]}{\exp\left(\lambda C_1\frac{n}{2}\right)}$$

$$= \inf_{\lambda\geq 0} \frac{\prod_{i=1}^{n^2\exp(-cn\bar{p})} \mathbb{E}\left[\exp(\lambda X_i)\right]}{\exp\left(\lambda C_1\frac{n}{2}\right)}$$

$$= \inf_{\lambda\geq 0} \frac{\prod_{i=1}^{n^2\exp(-cn\bar{p})} \left((e^\lambda-1)\bar{p}+1\right)}{\exp\left(\lambda C_1\frac{n}{2}\right)}$$

$$\overset{(a)}{\leq} \inf_{\lambda\geq 0} \frac{\exp\left(n^2\exp(-cn\bar{p})(e^\lambda-1)\bar{p}\right)}{\exp\left(\lambda C_1\frac{n}{2}\right)}$$

$$\leq \frac{\exp\left(n\cdot n\bar{p}\exp(-Cn\bar{p})(e^5-1)\right)}{\exp\left(C_1\frac{5n}{2}\right)}$$

$$\overset{(b)}{\leq} \exp(-2C_1 n)\,,$$

where $(a)$ and $(b)$ are from the inequality $1+x\leq e^x \;\forall x\in\mathbb{R}$ and the fact that $x\exp(-Cx)=o(1)$ when $x=\omega(1)$, respectively. Thus, we have

$$\mathbb{P}\left\{ \sum_{(v,w)\in\Gamma^c\times\mathcal{I}} A_{vw}^\ell \geq C_1 n \right\} \leq \exp(-2C_1 n)+\exp(-\omega(n))$$

$$\leq \exp(-C_1 n)\,.$$

Note that $|x_v y_w|\leq\sqrt{\frac{\bar{p}}{n}}$ for all $(v,w)\in\mathcal{L}$. Therefore, with probability at least $1-\exp(-C_1 n)$, we have

$$P_1(x,y) = \left| \sum_{(v,w)\in\mathcal{K}\cap\mathcal{L}} x_v A_{vw}^\ell y_w \right|$$

$$\leq \sum_{(v,w)\in\mathcal{K}\cap\mathcal{L}} A_{vw}^\ell |x_v y_w|$$

$$\leq 2\sqrt{\frac{\bar{p}}{n}} \sum_{(v,w)\in(\Gamma^c\times\mathcal{I})\cap\mathcal{L}} A_{vw}^\ell$$

$$\leq 2C_1\sqrt{n\bar{p}}\,.$$

**Bound on $P_2(x,y)$:** Using $\lambda=\frac{1}{2}\sqrt{\frac{n}{\bar{p}}}$, a positive constant $C$, and $\beta_{vw}:=x_v y_w\mathbb{1}\left\{(v,w)\in\mathcal{L}\right\}+x_w y_v\mathbb{1}\left\{(w,v)\in\mathcal{L}\right\}$ for all $(v,w)\in\mathcal{I}\times\mathcal{I}$, we have

$$\mathbb{P}\left\{ \sum_{(v,w)\in\mathcal{L}} x_v A_{vw}^\ell y_w - x^\top M^\ell y \geq C\sqrt{n\bar{p}} \right\} \leq \frac{\mathbb{E}\left[\exp\left(\lambda\left(\sum_{(v,w)\in\mathcal{I}\times\mathcal{I}: v>w} A_{vw}^\ell\beta_{vw}\right)\right)\right]}{\exp\left(\lambda\left(C\sqrt{n\bar{p}}+x^\top My\right)\right)}$$

$$= \frac{\prod_{(v,w)\in\mathcal{I}\times\mathcal{I}: v>w}\left(M_{vw}^\ell\left(\exp(\lambda\beta_{vw})-1\right)+1\right)}{\exp\left(\lambda\left(C\sqrt{n\bar{p}}+x^\top M^\ell y\right)\right)}$$

$$\leq \frac{\prod_{(v,w)\in\mathcal{I}\times\mathcal{I}: v>w}\exp\left(M_{vw}^\ell\left(\exp(\lambda\beta_{vw})-1\right)\right)}{\exp\left(\lambda\left(C\sqrt{n\bar{p}}+x^\top M^\ell y\right)\right)}$$

$$\leq \frac{\prod_{(v,w)\in\mathcal{I}\times\mathcal{I}:v>w} \exp\left(M_{vw}^{\ell}\left(\lambda\beta_{vw} + 2(\lambda\beta_{vw})^2\right)\right)}{\exp\left(\lambda\left(C\sqrt{n\overline{p}} + x^\top My\right)\right)}$$

$$= \frac{\exp\left(\sum_{(v,w)\in\mathcal{I}\times\mathcal{I}:v>w} M_{vw}^{\ell}\left(\lambda\beta_{vw} + 2(\lambda\beta_{vw})^2\right)\right)}{\exp\left(\lambda\left(C\sqrt{n\overline{p}} + x^\top M^\ell y\right)\right)}$$

$$= \exp\underbrace{\left(\sum_{(v,w)\in\mathcal{I}\times\mathcal{I}:v>w} M_{vw}^{\ell}\lambda\beta_{vw} - \lambda x^\top M^\ell y\right)}_{\text{(i)}}$$

$$\cdot \exp\underbrace{\left(2\sum_{(v,w)\in\mathcal{I}\times\mathcal{I}:v>w} M_{vw}^{\ell}(\lambda\beta_{vw})^2\right)}_{\text{(ii)}} \cdot \exp\left(-\lambda C\sqrt{n\overline{p}}\right) \qquad (25)$$

For (i) and (ii), we have following bounds.

(i):  This term can be alternatively expressed as follows:

$$\exp\left(\sum_{(v,w)\in\mathcal{I}\times\mathcal{I}:v>w} M_{vw}^{\ell}\lambda\beta_{vw} - \lambda x^\top M^\ell y\right) = \exp\left(\sum_{(v,w)\in\mathcal{L}} M_{vw}^{\ell}\lambda x_v y_w - \sum_{(v,w)\in\mathcal{I}\times\mathcal{I}} M_{vw}^{\ell}\lambda x_v y_w\right) \qquad (26)$$

$$= \exp\left(-\sum_{(v,w)\in\mathcal{L}^c} \lambda x_v y_w M_{vw}^{\ell}\right) . \qquad (27)$$

Note that $|x_v y_w| > \sqrt{\frac{\overline{p}}{n}}$ for all $(v,w)\in\mathcal{L}^c$ and $\sum_{(u,w)\in\mathcal{I}\times\mathcal{I}} x_u^2 y_w^2 = 1$. Therefore,

$$\sqrt{\frac{\overline{p}}{n}}\sum_{(u,v)\in\mathcal{L}^c} |x_v y_w| \leq \sum_{(u,v)\in\mathcal{L}^c} x_v^2 y_w^2$$

$$\leq 1 .$$

It implies

$$\sum_{(u,v)\in\mathcal{L}^c} |x_v y_w| \leq \sqrt{\frac{n}{\overline{p}}} .$$

Thus,

$$\left|\sum_{(v,w)\in\mathcal{L}^c} \lambda x_v y_w M_{vw}^{\ell}\right| \leq \sum_{(v,w)\in\mathcal{L}^c} \lambda |x_v y_w| M_{vw}^{\ell}$$

$$\leq \sum_{(v,w)\in\mathcal{L}^c} \lambda |x_v y_w| \overline{p}$$

$$= \frac{1}{2}\sqrt{n\overline{p}}\sum_{(v,w)\in\mathcal{L}^c} |x_v y_w|$$

$$\leq \frac{n}{2} . \qquad (28)$$

From (27) and (28), we have

$$\exp\left(\sum_{(v,w)\in\mathcal{I}\times\mathcal{I}:v>w} M_{vw}^{\ell}\lambda\beta_{vw} - \lambda x^\top M^\ell y\right) \leq \exp\left(\frac{n}{2}\right) .$$

(ii): Note that from the definition,

$$|\beta_{vw}|^2 \leq 2(x_v y_w)^2 \mathbb{1}\{(v,w) \in \mathcal{L}\} + 2(x_w y_v)^2 \mathbb{1}\{(w,v) \in \mathcal{L}\}$$

holds. Thus, we have

$$2 \sum_{(v,w) \in \mathcal{I} \times \mathcal{I}: v > w} M_{vw}^\ell (\lambda \beta_{vw})^2$$

$$\leq 2 \sum_{(v,w) \in \mathcal{I} \times \mathcal{I}: v > w} \bar{p} \cdot \frac{n}{4\bar{p}} \cdot \beta_{vw}^2$$

$$= \frac{n}{2} \sum_{(v,w) \in \mathcal{I} \times \mathcal{I}: v > w} \beta_{vw}^2$$

$$\leq \frac{n}{2} \sum_{(v,w) \in \mathcal{I} \times \mathcal{I}: v > w} \left(2(x_v y_w)^2 \mathbb{1}\{(v,w) \in \mathcal{L}\} + 2(x_w y_v)^2 \mathbb{1}\{(w,v) \in \mathcal{L}\}\right)$$

$$= n \sum_{(v,w) \in \mathcal{L}} (x_v y_w)^2$$

$$\overset{(a)}{\leq} n \ ,$$

where in $(a)$ we used $\sum_{(v,w) \in \mathcal{L}} (x_v y_w)^2 \leq \sum_{(v,w) \in \mathcal{I} \times \mathcal{I}} (x_v y_w)^2 = 1$. Taking exponential of the previous inequalities, we have:

$$\exp\left(2 \sum_{(v,w) \in \mathcal{I} \times \mathcal{I}: v > w} M_{vw}^\ell (\lambda \beta_{vw})^2\right) \leq \exp(n)$$

Combining the bounds on (i) and (ii) with (25), we have:

$$\mathbb{P}\left\{\sum_{(v,w) \in \mathcal{L}} x_v A_{vw}^\ell y_w - x^\top M^\ell y \geq C\sqrt{n\bar{p}}\right\} \leq \exp\left(\frac{3-C}{2}n\right) \ .$$

Taking any $C_2 > 3$, we have:

$$\sum_{(v,w) \in \mathcal{L}} x_v A_{vw}^\ell y_w - x^\top M^\ell y \leq C_2 \sqrt{n\bar{p}},$$

with probability at least $1 - \exp\left(-\frac{C_2 - 3}{2}n\right)$.

**Bound on $P_3(x,y)$:** We extend the proofs in (Feige & Ofek, 2005). For any $A, B \subset \mathcal{I}$, let we define $e(A,B) := \sum_{v \in A} \sum_{w \in B} A_{vw}^\ell$ to be the number of positive labels between the nodes in $A$ and the nodes in $B$ and $\mu(A,B) := |A||B|\bar{p}$. $\mu(A,B)$ is an upper bound of the expected number of labels between $A$ and $B$. We call the adjacency matrix $A_\Gamma^\ell$ has the *bounded degree property* with a positive constant $c_1 > 0$ if for all $v \in \mathcal{I} \cap \Gamma$, $e(v, \mathcal{I} \cap \Gamma) \leq c_1 n\bar{p}$ holds. Furthermore, we state that the adjacency matrix $A_\Gamma^\ell$ has the *discrepancy property* with constants $c_2 > 0$ and $c_3 > 0$ if for every $A, B \subset \mathcal{I} \cap \Gamma$, one of the following holds:

$$\text{(i)} \quad \frac{e(A,B)}{\mu(A,B)} \leq c_2 \tag{29}$$

$$\text{(ii)} \quad e(A,B) \log\left(\frac{e(A,B)}{\mu(A,B)}\right) \leq c_3 |B| \log\left(\frac{n}{|B|}\right) \ . \tag{30}$$

By Corollary 2.11 in (Feige & Ofek, 2005), if the graph with the adjacency matrix $A_\Gamma$ satisfies the discrepancy and the bounded degree properties, there exists a constant $C'$ which depends on $c_1$, $c_2$, and $c_3$ such that

$$P_3(x,y) \leq C' \sqrt{n\bar{p}},$$

for all $x, y \in T$.

First, we have the following lemma that guarantees the probability that the bounded degree property holds.

**Lemma E.1.** *For any constant $C > 0$, there exists a constant $c_1 > 0$ such that the bounded degree property of $A_\Gamma$ with $c_1$ holds with probability at least $1 - \exp(-Cn\bar{p})$.*

The proof is given in Appendix E.8.

Next, we have the following lemma that guarantees the probability that the discrepancy property holds.

**Lemma E.2.** *For any $C > 0$, there are positive constants $c_2$ and $c_3$ such that the discrepancy property holds with probability at least $1 - \exp(-Cn\bar{p})$.*

The proof of Lemma E.2 is given in Appendix E.9. Therefore, for any $C > 0$, there exists $c_1$, $c_2$, and $c_3$ such that the bounded degree property and discrepancy property hold with probability at least $1 - \exp(-Cn\bar{p})$.

**Bound on $P_4(x, y)$:** We have, with a constant $C > e^L$, for all $x, y \in T$, with probability at least $1 - \exp(-\omega(n))$,

$$
\begin{aligned}
P_4(x, y) = |x^\top (M^\ell - M_\Gamma^\ell) y| \\
\leq \|M^\ell - M_\Gamma^\ell\|_2 \\
\leq \|M^\ell - M_\Gamma^\ell\|_F \\
\leq \sqrt{\sum_{(v,w)\in\mathcal{K}} \bar{p}^2} \\
= \sqrt{(\lfloor n\exp(-n\widetilde{p})\rfloor)^2 \bar{p}^2} \\
\overset{(a)}{\leq} \sqrt{(n\exp(-Cn\bar{p}))^2 \bar{p}^2} \\
= n\bar{p}\exp(-Cn\bar{p})
\end{aligned}
$$

where we used Lemma C.5 in $(a)$. This concludes the proof. ∎

### E.6. Proof of Lemma 4.3

For all $v, w \in \Gamma$ such that $\sigma(v) \neq \sigma(w)$, there exists a constant $C > 0$ that depends on $\varepsilon$ in (A2) such that with probability at least $1 - \exp(-\omega(n))$,

$$
\begin{aligned}
\|M_{\Gamma,v} - M_{\Gamma,w}\|_2^2 = \sum_{i\in\mathcal{I}\cap\Gamma} \sum_{\ell=1}^{L} (p(\sigma(v), \sigma(i), \ell) - p(\sigma(w), \sigma(i), \ell))^2 \\
= \sum_{k\in[K]} \sum_{i\in\mathcal{I}_k\cap\Gamma} \sum_{\ell=1}^{L} (p(\sigma(v), \sigma(i), \ell) - p(\sigma(w), \sigma(i), \ell))^2 \\
\overset{(a)}{\geq} \sum_{k\in[K]} (\alpha_k - \exp(-C_1 n\bar{p})) n \sum_{\ell=1}^{L} (p(\sigma(v), \sigma(i), \ell) - p(\sigma(w), \sigma(i), \ell))^2 \\
\overset{(b)}{\geq} C\varepsilon n\bar{p}^2,
\end{aligned}
$$

where for $(a)$, we used Lemma C.5 to replace $\widetilde{p}$ with $\bar{p}$ in the cardinality of $\Gamma$; and for $(b)$, we used (A2).

∎

### E.7. Proof of Lemma 4.4

First, from Lemma 4.2, for any $c > 0$, there exists constants $C_1, C_2 > 0$ such that $\sigma_K \geq C_1 n\bar{p}$ and $\sigma_{K+1} \leq C_2\sqrt{n\bar{p}}$ with probability at least $1 - \exp(-cn\bar{p})$. Therefore, for any $c > 0$, we have $\hat{K} = K$ with probability at least $1 - \exp(-cn\bar{p}(1 + o(1)))$. From the analysis of the iterative power method in (Halko et al., 2011) (Theorem 9.2 and Theorem 9.1 in (Halko et al., 2011)), for any $c > 0$, with probability at least $1 - \exp(-cn\bar{p}(1 + o(1)))$,

$$
\|\bar{A} - \hat{A}\|_2 \leq (1 + \|\Omega_2 \Omega_1^{-1}\|_2^2)^{\frac{1}{4\lceil(\log n)^2\rceil+2}} \sigma_{K+1},
$$

where $\Omega_1$ is a $K \times K$ standard Gaussian random matrix, $\Omega_2$ is an $Ln - K \times K$ Gaussian random matrix, $\sigma_{K+1}$ is the $K + 1$-th largest singular value of the matrix $A_\Gamma$. We use the following proposition from (Halko et al., 2011).

**Proposition E.3** (Proposition A.3. in (Halko et al., 2011))**.** *Let $G$ be a $K \times K$ standard Gaussian matrix with $K \geq 2$. For each $t > 0$,*

$$\mathbb{P}(\|G^{-1}\|_2 > t) \leq e\sqrt{\frac{K}{2\pi}}\frac{1}{t}.$$

From this proposition, for any $c > 0$, there exists a constant $C > 0$ such that $\|\Omega_1^{-1}\|_2 \leq Cn^c$ with probability at least $1 - 1/n^c$.

**Theorem E.4** (Theorem 4.4.5 of (Vershynin, 2018).)**.** *Let $A$ be an $m \times n$ random matrix whose entries are independent standard Gaussian random variables. For any $t > 0$, we have*

$$\|A\|_2 \leq C(\sqrt{n} + \sqrt{m} + t),$$

*with probability at least $1 - 2\exp(-t^2)$, where $C > 0$ is some constant.*

Therefore, for any $c > 0$, there exist constants $C_1, C_2 > 0$ such that

$$\begin{aligned}
\|\bar{A} - \hat{A}\|_2 &\leq (1 + \|\Omega_2\|_2^2\|\Omega_1^{-1}\|_2^2)^{\frac{1}{4\lceil(\log n)^2\rceil+2}}\sigma_{K+1} \\
&\leq (1 + Cn \cdot n^c)^{\frac{1}{4\lceil(\log n)^2\rceil+2}}\sigma_{K+1} \\
&\leq C_2\sigma_{K+1},
\end{aligned}$$

with probability at least $1 - 1/n^c$. This concludes the proof. ∎

### E.8. Proof of Lemma E.1

Recall that from Lemma E.1, for each $v \in \mathcal{I}$, for any $C > 0$, we have

$$e(v, \mathcal{I}) \leq Cn\bar{p}, \tag{31}$$

with probability at least $\exp(-(C - e^L)n\bar{p})$. Also, from Lemma C.5, for any constant $C_2 > e^L$, we have $\tilde{p} \leq C_2\bar{p}$ with probability at least $1 - \exp(-\omega(n))$. The number of trimmed nodes is larger than $n\exp(-C_2n\bar{p})$ with probability at least $1 - \exp(-\omega(n))$. With these $C$ and $C_2$, by Markov's inequality,

$$\begin{aligned}
\mathbb{P}\left\{\text{The number of nodes that do not satisfy (31)} \geq n\exp(-C_2n\bar{p})\right\} &\leq \frac{\mathbb{E}\left[\sum_{v \in \mathcal{I}}\mathbb{1}\left\{e(v, \mathcal{I}) > C'n\bar{p}\right\}\right]}{n\exp(-C_2n\bar{p})} \\
&= \frac{\sum_{v \in \mathcal{I}}\mathbb{P}\left\{e(v, \mathcal{I}) > C'n\bar{p}\right\}}{n\exp(-C_2n\bar{p})} \\
&\leq \exp\left(-(C - C_2 - e^L)n\bar{p}\right).
\end{aligned}$$

Therefore, by the union bound, for any $C > 0$, there exists a constant $C' > 0$ such that any $v \in \mathcal{I} \cap \Gamma$ satisfies $e(v, \mathcal{I}) \leq C'n\bar{p}$, with probability at least $1 - \exp(-Cn\bar{p}) - \exp(-\omega(n))$. This concludes the proof.

### E.9. Proof of Lemma E.2

Let $A$ and $B$ be subsets of $\mathcal{I} \cap \Gamma$. Without loss of generality, we assume that $|A| \leq |B|$. We prove the lemma by dividing it into two cases.

**Case 1:** when $|B| \geq n/5$. By Lemma E.1, for any constant $C > 0$, there exists a constant $c_1$ such that for all $v \in \mathcal{I} \cap \Gamma$, $e(v, \mathcal{I} \cap \Gamma) \leq c_1n\bar{p}$ holds with probability at least $1 - \exp(-Cn\bar{p})$. Therefore, for any constant $C > 0$, there exists a constant $c_2 > 0$ such that with probability at least $1 - \exp(-Cn\bar{p})$ for all $A, B \subset \mathcal{I} \cap \Gamma$ such that $|B| \geq |A|$,

$$e(A, B) \leq e(A, \mathcal{I} \cap \Gamma)$$

$$\leq |A|c_1 n\bar{p}$$

$$= 5c_1|A|\frac{n}{5}\bar{p}$$

$$\leq 5c_1|A||B|\bar{p}$$

$$\overset{(a)}{\leq} c_2\mu(A, B),$$

where in $(a)$ we put $c_2 \geq 5c_1$.

**Case 2:** when $|B| \leq n/5$. Let we define $\eta(A, B) = \max(\eta^0, c_2)$ where $\eta^0$ is the solution that satisfies $\eta^0\mu(A, B)\log\eta^0 = c_3|B|\log(n/|B|)$. As $c_3|B|\log(n/|B|) > 0$, it implies $\eta^0 > 1$. Furthermore, as $x\log x$ is an strictly increasing function when $x \geq 1$, $\eta^0$ is unique for fixed $A$ and $B$.

When $e(A, B) \leq \eta(A, B)\mu(A, B)$ and $\eta(A, B) = c_2$, the condition (i) is satisfied. When $e(A, B) \leq \eta(A, B)\mu(A, B)$ and $\eta(A, B) = \eta^0$, using the definition of $\eta^0$, we have

$$e(A, B) \leq \eta^0\mu(A, B)$$
$$= \frac{1}{\log\eta^0}c_3|B|\log\left(\frac{n}{|B|}\right) .$$

From $\log(e(A, B)/\mu(A, B)) \leq \log\eta^0$, we have

$$e(A, B)\log\left(\frac{e(A, B)}{\mu(A, B)}\right) \leq c_3|B|\log\left(\frac{n}{|B|}\right) .$$

Hence, the condition (ii) is satisfied. Therefore, when $e(A, B) \leq \eta(A, B)\mu(A, B)$, the discrepancy property holds. We quantifies the probability that $e(A, B) \leq \eta(A, B)\mu(A, B)$ holds for any $A, B \subset \mathcal{I} \cap \Gamma$.

First, by Markov's inequality,

$$\mathbb{P}(e(A, B) > \eta(A, B)\mu(A, B)) \leq \inf_{\lambda \geq 0} \frac{\mathbb{E}[\exp(\lambda e(A, B))]}{\exp(\lambda\eta(A, B)\mu(A, B))}$$

$$\leq \inf_{\lambda \geq 0} \frac{(\bar{p}(\exp(\lambda) - 1) + 1)^{|A||B|}}{\exp(\lambda\eta(A, B)\mu(A, B))}$$

$$\leq \inf_{\lambda \geq 0} \frac{(\bar{p}\exp(\lambda) + 1)^{|A||B|}}{\exp(\lambda\eta(A, B)\mu(A, B))}$$

$$\leq \inf_{\lambda \geq 0} \frac{\exp(|A||B|\bar{p}\exp(\lambda))}{\exp(\lambda\eta(A, B)\mu(A, B))}$$

$$= \inf_{\lambda \geq 0} \exp\left(\exp(\lambda)\mu(A, B) - \lambda\eta(A, B)\mu(A, B)\right)$$

$$\overset{(a)}{\leq} \exp\left(-\eta(A, B)\mu(A, B)\left(\log\eta(A, B) - 1\right)\right),$$

where in $(a)$, we set $\lambda = \log\eta(A, B)$.

Next, we compute the probability as follows.

$$\mathbb{P}\left\{e(A, B) > \eta(A, B)\mu(A, B) \text{ for some } A, B \in \mathcal{I} \cap \Gamma, |B| \leq \frac{n}{5}, |A| \leq |B|\right\}$$

$$= \mathbb{P}\left\{\cup_{b=1}^{n/5} \cup_{a=1}^{b} \cup_{A, B \subset \mathcal{I} \cap \Gamma : |A| = a, |B| = b} \{e(A, B) > \eta(A, B)\}\right\}$$

$$\overset{(a)}{\leq} \sum_{b=1}^{n/5}\sum_{a=1}^{b} \binom{n}{a}\binom{n}{b} \exp\left(-\eta(A, B)\mu(A, B)\left(\log\eta(A, B) - 1\right)\right)$$

$$\overset{(b)}{\leq} \sum_{b=1}^{n/5}\sum_{a=1}^{b} \left(\frac{ne}{b}\right)^{2b} \exp\left(-\eta(A, B)\mu(A, B)\left(\log\eta(A, B) - 1\right)\right)$$

$$= \sum_{b=1}^{n/5} \sum_{a=1}^{b} \exp\left(2b + 2b\log\left(\frac{n}{b}\right) - \eta(A,B)\mu(A,B)\left(\log\eta(A,B) - 1\right)\right)$$

$$\overset{(c)}{\leq} \sum_{b=1}^{n/5} \sum_{a=1}^{b} \exp\left(4b\log\left(\frac{n}{b}\right) - \eta(A,B)\mu(A,B)\left(\log\eta(A,B) - 1\right)\right)$$

$$= \sum_{b=1}^{n/5} \sum_{a=1}^{b} \exp\left(-(C+2)\log n + (C+2)\log n + 4b\log\left(\frac{n}{b}\right) - \eta(A,B)\mu(A,B)\left(\log\eta(A,B) - 1\right)\right)$$

$$\overset{(d)}{\leq} \sum_{b=1}^{n/5} \sum_{a=1}^{b} \exp\left(-(C+2)\log n + (C+6)b\log\left(\frac{n}{b}\right) - \eta(A,B)\mu(A,B)\left(\log\eta(A,B) - 1\right)\right)$$

$$\overset{(e)}{\leq} \sum_{b=1}^{n/5} \sum_{a=1}^{b} \exp\left(-(C+2)\log n + (C+6)b\log\left(\frac{n}{b}\right) - \frac{\eta(A,B)\mu(A,B)}{2}\log\eta(A,B)\right)$$

$$\overset{(f)}{\leq} \sum_{b=1}^{n/5} \sum_{a=1}^{b} \exp\left(-(C+2)\log n + (C+6)b\log\left(\frac{n}{b}\right) - \frac{\eta^0\mu(A,B)}{2}\log\eta^0\right)$$

$$\overset{(g)}{\leq} \sum_{b=1}^{n/5} \sum_{a=1}^{b} \exp\left(-(C+2)\log n\right)$$

$$\leq \frac{1}{n^C},$$

where for $(a)$ is from the union bound; for $(b)$, we used $b \leq n/5$ and $a \leq b$; for $(c)$, we again used $b \leq n/5$; for $(d)$, we used the fact that $x\log(n/x)$ is an increasing function in $[1, n/e]$ (Lemma 2.12 in (Feige & Ofek, 2005)); for $(e)$, we used $\eta(A,B) \geq c_2$ with sufficiently large constant $c_2$; for $(f)$, we used the fact that $x\log x$ is a strictly increasing function when $x \geq 1$; $(g)$ stems from the definition of $\eta^0$ with $c_3 \geq 2(C+6)$.

From the assumption $\bar{p} = \mathcal{O}(\log n/n)$, for any $C > 0$, there exist constants $c_2$ and $c_3$ such that the discrepancy property with constants $c_2$ and $c_3$ holds with probability at least $1 - \exp(-Cn\bar{p})$. This concludes the proof. ∎

### E.10. Proof of Corollary 2.1

Let $\varepsilon^{\text{init}}(n)$ represent the number of misclassified nodes in the initial spectral clustering $\text{USC}(\tau)$ as described in (Gao et al., 2017). Based on the node-wise refinement guarantee provided in the proof of Theorem 4 in (Gao et al., 2017) (specifically, Lemma 17 of (Gao et al., 2017)), when $K = \mathcal{O}(1)$,

$$\sup_{(p,\alpha)\in\Theta(n,a,b)} \max_{i\in\mathcal{I}} \mathbb{P}_{(p,\alpha)}(\text{node } i \text{ is misclassified})$$

$$\leq \underbrace{\exp\left(-(1+o(1))\frac{nI^*}{K}\right)}_{(a)} + C\mathbb{P}_{(p,\alpha)}(\varepsilon^{\text{init}}(n) \leq \gamma_n), \tag{32}$$

where $\gamma_n$ is a positive sequence that satisfies $\lim_{n\to\infty} \gamma_n = 0$. According to Theorem 6 in (Gao et al., 2017), for any $C' > 0$, we obtain

$$\mathbb{P}_{(p,\alpha)}(\varepsilon^{\text{init}}(n) \leq \gamma_n) \leq \frac{1}{n^{C'}},$$

with an appropriate choice of the sequence $\gamma_n$ in (32). Since $I^* \asymp \frac{(a-b)^2}{na}$ according to (Gao et al., 2017), using the assumptions $a/b = \Theta(1)$ and $a = \mathcal{O}(\log n)$, we obtain

$$(a) \geq \exp\left(-C''\log n\right) = \frac{1}{n^{C''}},$$

for some constant $C'' > 0$. Therefore, since we can choose any constant $C' > 0$,

$$C\mathbb{P}_{(p,\alpha)}(\varepsilon^{\text{init}}(n) \leq \gamma_n) = o\left(\exp\left(-(1+o(1))\frac{nI^*}{K}\right)\right)$$

and

$$\sup_{(p,\alpha)\in\Theta(n,a,b)} \max_{i\in\mathcal{I}} \mathbb{P}_{(p,\alpha)}(\text{node } i \text{ is misclassified}) \leq \exp\left(-(1+o(1))\frac{nI^*}{K}\right).$$

Thus, we obtain

$$\begin{aligned}
\sup_{(p,\alpha)\in\Theta(n,a,b)} \mathbb{E}_{(\alpha,p)}[\varepsilon^{\text{PLMLE}}(n)] &= \sup_{(p,\alpha)\in\Theta(n,a,b)} \sum_{i\in\mathcal{I}} \mathbb{P}_{(p,\alpha)}(\text{node } i \text{ is misclassified}) \\
&\leq \sup_{(p,\alpha)\in\Theta(n,a,b)} n \max_{i\in\mathcal{I}} \mathbb{P}_{(p,\alpha)}(\text{node } i \text{ is misclassified}) \\
&\leq n \exp\left(-(1+o(1))\frac{nI^*}{K}\right).
\end{aligned}$$

This concludes the proof. ∎

## F. Numerical Experiments

In this section, we present numerical evaluations of the proposed algorithm. Our experiments are based on the code of (Wang et al., 2021), and we consider the three scenarios proposed in (Gao et al., 2017), as well as an additional scenario. The main focus of our comparison is the IAC algorithm (Algorithm 1) and a computationally efficient version of the penalized local maximum likelihood estimation (PLMLE) algorithm (Algorithm 3 in (Gao et al., 2017)). Note that this version of PLMLE does not have performance guarantees, but it still requires $\Omega(n^2)$ floating-point operations. As shown in Section 4 of (Gao et al., 2017), PLMLE and its simplified version (Algorithm 3) exhibit nearly identical performance in all considered scenarios. In all experiments, we consider simple SBMs with $L = 1$.

*Table 2.* Number of misclassified items. IAC and PLMLE indicate Algorithm 1 and Algorithm 3 in (Gao et al., 2017), respectively. Means and standard deviations are calculated from the results of 100 experiment instances.

| Model | Algorithm | Mean | Std |
|-------|-----------|------|-----|
| Model 1 | IAC | 2.8800 | 1.5909 |
| | PLMLE | 2.9700 | 1.6542 |
| Model 2 | IAC | 0.0000 | 0.0000 |
| | PLMLE | 0.1850 | 0.4262 |
| Model 3 | IAC | 29.4100 | 4.9789 |
| | PLMLE | 31.0400 | 5.1775 |
| Model 4 | IAC | 45.5600 | 9.2489 |
| | PLMLE | 54.7400 | 10.5329 |

**Model 1: Balanced Symmetric.** First, consider the SBM corresponding to the "Balanced case" in (Gao et al., 2017). Assume that $n = 2500$, $K = 10$, and $L = 1$. We fix the community size to be equal as $\forall k \in [10], |\mathcal{I}_k| = 250$. We set the observation probability as $p(k,k,1) = 0.48$ for all $k$ and $p(i,k,1) = 0.32$ for all $i \neq k$.

**Model 2: Imbalanced.** The next SBM corresponds to the "Imbalanced case" in (Gao et al., 2017). We set $n = 2000$, $K = 4$, and $L = 1$. The sizes of the clusters are heterogenous: $|\mathcal{I}_1| = 200$, $|\mathcal{I}_2| = 400$, $|\mathcal{I}_3| = 600$, and $|\mathcal{I}_4| = 800$. We set the observation probability matrix $(p(i,k,1))_{i,k}$ as in (Gao et al., 2017):

$$(p(i,k,1))_{i,k} = \begin{pmatrix} 0.50 & 0.29 & 0.35 & 0.25 \\ 0.29 & 0.45 & 0.25 & 0.30 \\ 0.35 & 0.25 & 0.50 & 0.35 \\ 0.25 & 0.30 & 0.35 & 0.45 \end{pmatrix}. \tag{33}$$

**Model 3: Sparse Symmetric.** The last experimental setting from (Gao et al., 2017) is the sparse and symmetric case. We generate networks with $n = 4000$, $K = 10$, and $L = 1$. Clusters are of equal sizes: $\forall k \in [10], |\mathcal{I}_k| = 400$. We set the statistical parameter as $p(k,k,1) = 0.032$ for all $k$ and $p(i,k,1) = 0.005$ for all $i \neq k$.

**Model 4: Sparse Asymmetric.** Lastly, we consider the cluster recovery problem with a sparse and asymmetric statistical parameter. We set $n = 1200$, $K = 4$, and $L = 1$. Clusters are of equal sizes: $\forall k \in [4], |\mathcal{I}_k| = 300$. We fix the statistical parameter $(p(i, k, 1))_{i,k}$ as

$$(p(i, k, 1))_{i,k} = \begin{pmatrix} 0.032 & 0.005 & 0.008 & 0.005 \\ 0.005 & 0.028 & 0.005 & 0.008 \\ 0.008 & 0.005 & 0.032 & 0.005 \\ 0.005 & 0.008 & 0.005 & 0.028 \end{pmatrix}.$$

The simulations presented in this paper were conducted using the following computational environment.

- Operating System: macOS Sonoma

- Programming Language: MATLAB

- Processor: Apple M3 Max

- Memory: 128 GB

The average running time of our IAC algorithm is $0.177 \pm 0.037$ seconds, while that of PLMLE is $0.915 \pm 0.752$ seconds for Model 4.

The results of our experiments are summarized in Table 2. The IAC algorithm consistently performs slightly better than Algorithm 3 in (Gao et al., 2017). Figures 1, 2, 3, and 4 display boxplots representing the number of misclassified nodes for each Model and method.

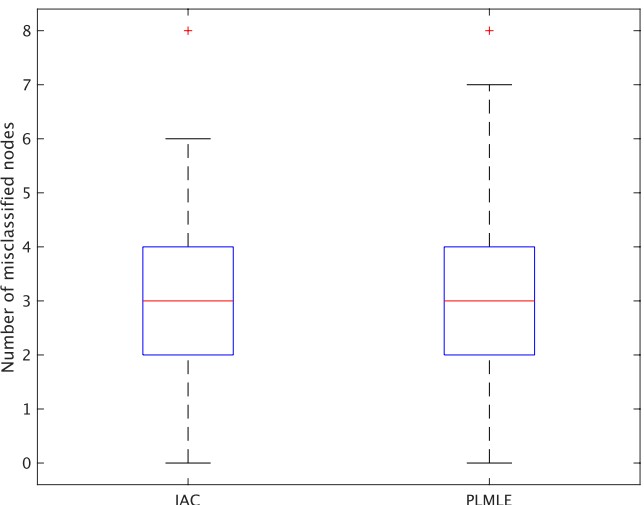

*Figure 1.* Number of misclassified nodes for the balanced symmetric case (Model 1). IAC and PLMLE indicate Algorithm 1 and Algorithm 3 in (Gao et al., 2017), respectively. The figure is plotted with MATLAB boxplot function with outliers (the red crosses) for 100 experiment instances.

### F.1. Experiments with Real-World Dataset

We applied our algorithm to a real-world dataset, the DBLP citation network dataset (Backstrom et al., 2006). In this dataset, the nodes represent researchers. We focused on 246 researchers who have authored 20 or more papers, selected from a corpus of 50,000 papers. The edges represent co-authorship relationships: two researchers are connected if they have co-authored at least one paper. There are 1,118 such co-authorship connections in the network.

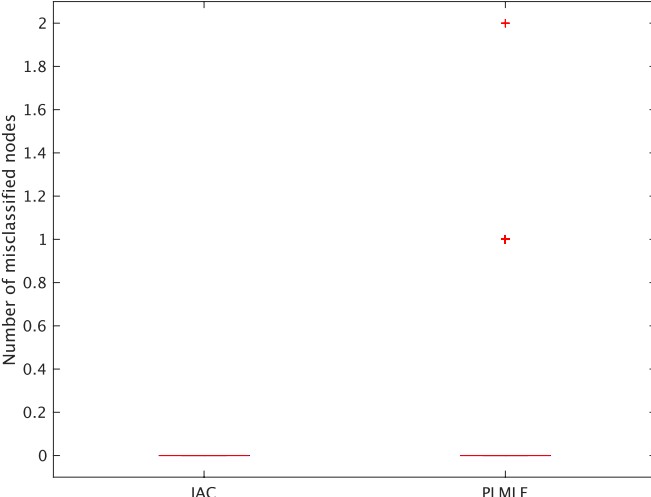

*Figure 2.* Number of misclassified nodes for the imbalanced case (Model 2). IAC and PLMLE indicate Algorithm 1 and Algorithm 3 in (Gao et al., 2017), respectively. The figure is plotted with MATLAB boxplot function with outliers (the red crosses) for 100 experiment instances.

The researchers in this network were clustered using both IAC and PLMLE, with the number of clusters set to 8 for simplicity. Network visualizations and statistics (community size distributions and the distributions of publications per author in each community) are shown in Figures 5 and 6, respectively. Interestingly, our IAC algorithm discovered larger communities without excessively small ones, resulting in a more continuous community size distribution.

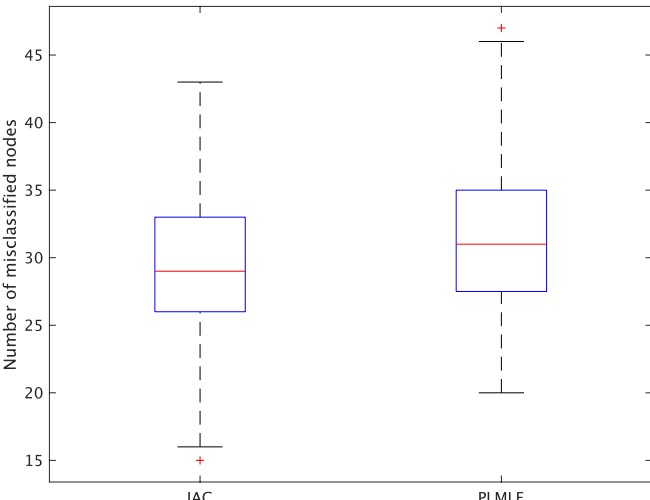

*Figure 3.* Number of misclassified nodes for the sparse symmetric case (Model 3). IAC and PLMLE indicate Algorithm 1 and Algorithm 3 in (Gao et al., 2017), respectively. The figure is plotted with MATLAB boxplot function with outliers (the red crosses) for 100 experiment instances.

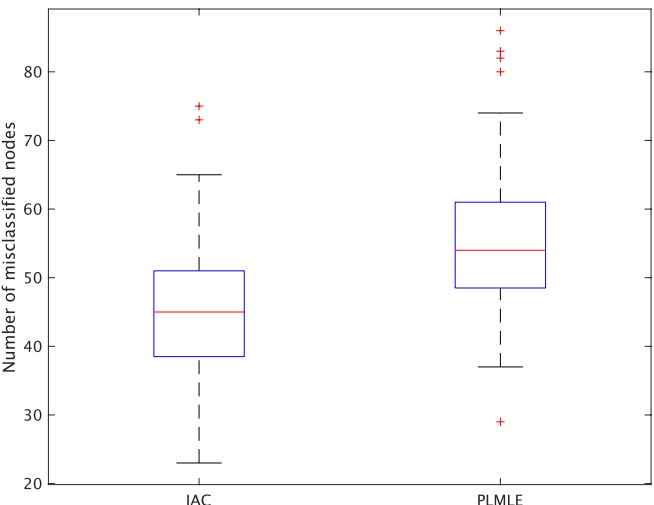

*Figure 4.* Number of misclassified nodes for the sparse asymmetric case (Model 4). IAC and PLMLE indicate Algorithm 1 and Algorithm 3 in (Gao et al., 2017), respectively. The figure is plotted with MATLAB boxplot function with outliers (the red crosses) for 100 experiment instances.

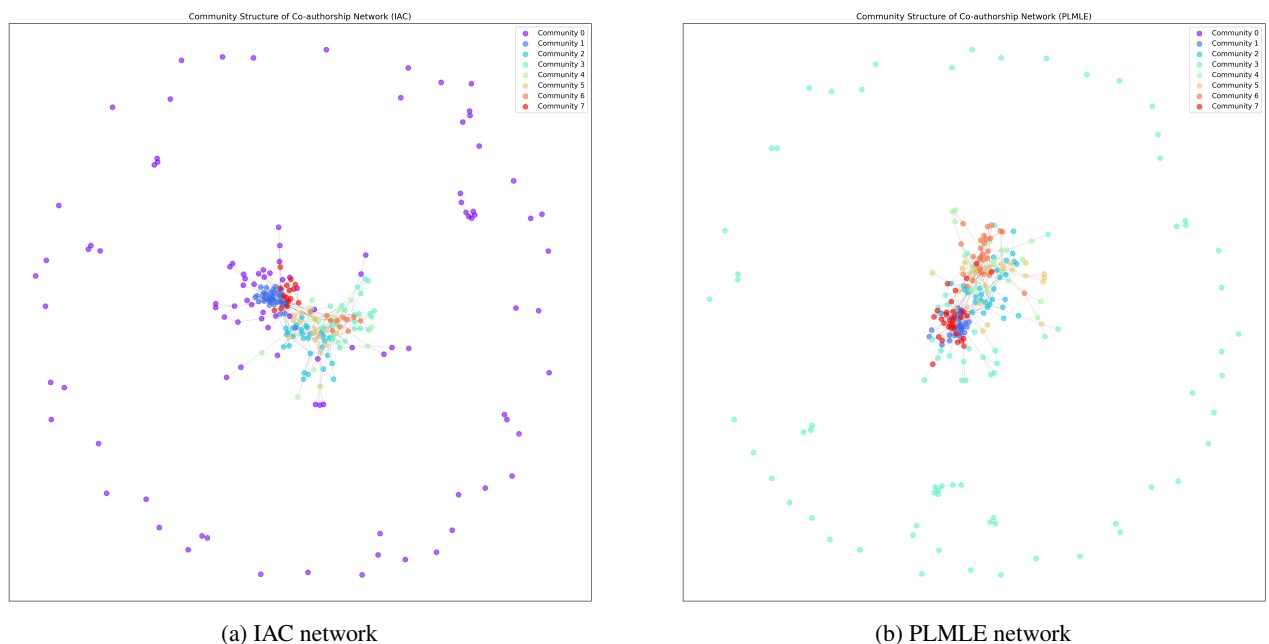

(a) IAC network          (b) PLMLE network

*Figure 5.* Visualization of clustered networks: (a) IAC, (b) PLMLE.

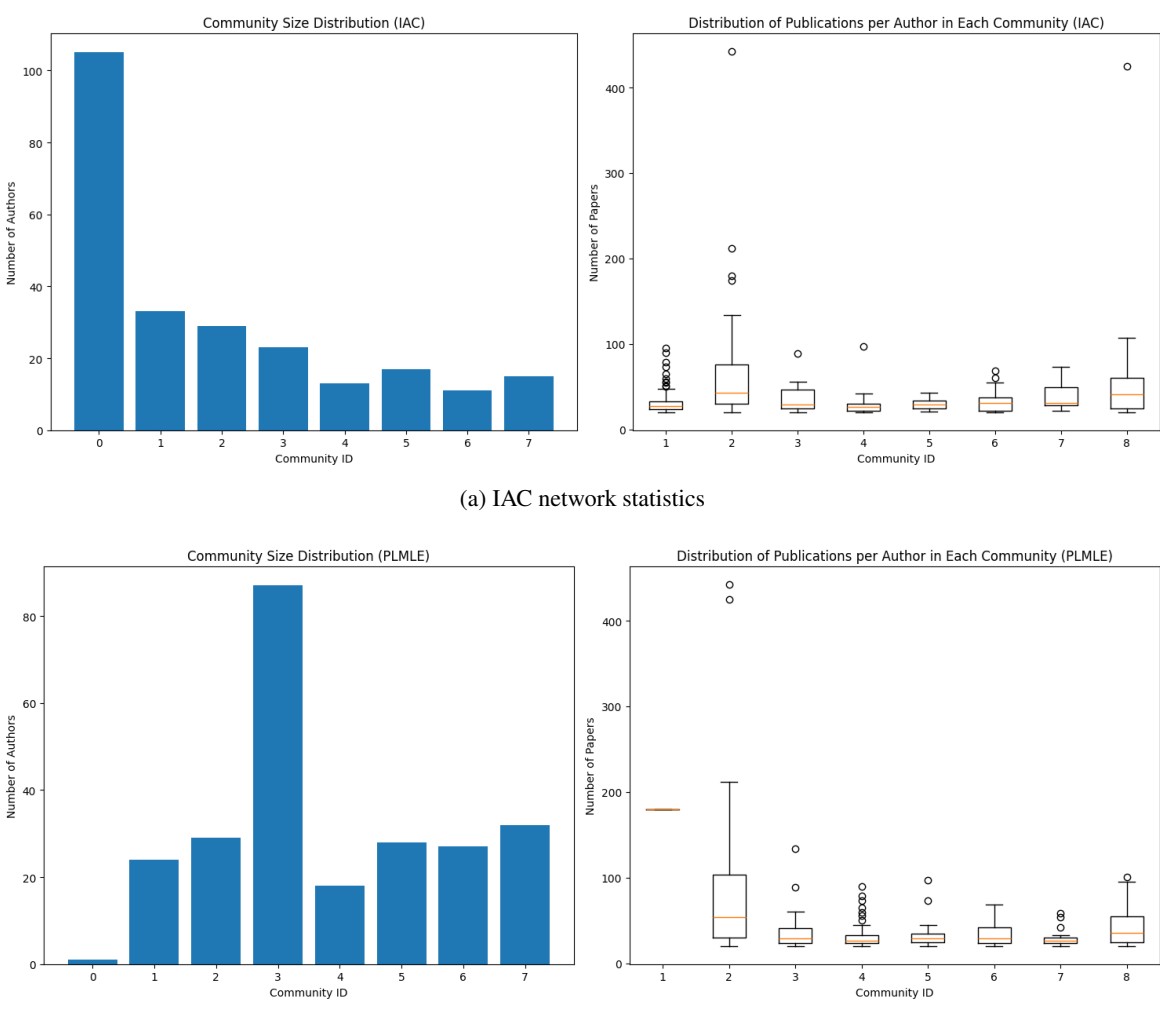

(a) IAC network statistics

(b) PLMLE network statistics

*Figure 6.* Statistics of the clustered networks: (a) IAC, (b) PLMLE.

