# OpenReview forum: "Revisiting Instance-Optimal Cluster Recovery in the Labeled Stochastic Block Model"
_ICML.cc/2025/Conference — ICML 2025 poster_

### Official Review · Reviewer_JfS5 · 2025-03-12

**Overall Recommendation:** 3

**Summary:**

The authors study the labeled stochastic block model, which is similar to the regular SBM, but each edge is additionally assigned one label from a candidate set. 0 corresponds to no edge existing, and the authors study the sparse regime where the probability of a label not being 0 is small ($O(\log n / n)$).
In this regime, the authors show that given some assumptions on the LSBM concerning homogeneity of the labels, cluster separation and the (non)existence of labels that are too sparse, there exists an instance optimal algorithm, meaning for any LBSM satisfying these assumptions the algorithm will misclassify a sublinear amount of vertices. This clustering algorithm matches a lower bound from the literature.

**Claims And Evidence:**

I can not judge the evidence as all proofs are in the appendix and I did not have time to look at them in detail.

**Essential References Not Discussed:**

Not aware of any.

**Experimental Designs Or Analyses:**

The experiments are performed for different regimes of the SBM, with both algorithms performing similar in the first three settings and the algorithm IAC performing better in the sparse asymmetric setting. IAC is also better in the first three settings, but the differences are very small. Additional data such as the running time would have been interesting to get a fuller picture on the practical properties of the algorithms.

**Methods And Evaluation Criteria:**

There are some experiments in the appendix, the authors compare their own algorithm to one algorithm from the literature over a benchmark dataset for various regimes of the SBM. The labeled SBM is not considered, likely due to the algorithm of Gao being for the regular SBM.

The results are evaluated based on the number of misclassified nodes.

**Other Comments Or Suggestions:**

None

**Other Strengths And Weaknesses:**

None

**Questions For Authors:**

None

**Relation To Broader Scientific Literature:**

The SBM is a standard model for studying the theoretical limitations of clustering algorithms. I am not familiar with the labelled SBM and wish the authors would have motivated the uses of the model a bit more. The results may inform researchers in other areas of clustering about the theoretical limitations of algorithms, but whether these results have implications for clustering on graphs in real-world instances is not entirely clear to me.

**Theoretical Claims:**

I did not check the proofs of the theoretical claims.

---

> ### Author Rebuttal · Authors · 2025-03-31
>
> Thank you very much for your positive feedback and for carefully reading our draft.
>
> > Additional data such as the running time would have been interesting to get a fuller picture on the practical properties of the algorithms.
>
> Thank you for this suggestion. We have conducted additional experiments to measure the running times of the algorithms. For Model 4, the average running time of our IAC algorithm is 0.177 ± 0.037 seconds, while that of PMLE is 0.915 ± 0.752 seconds.
> In the next revision, we will include these results, along with further scaling evaluations (by varying the model parameters and the sample size $n$), in the appendix of the paper.
>
> > Whether these results have implications for clustering on graphs in real-world instances is not entirely clear to me.
>
> We have applied our algorithm to a real-world dataset, the DBLP citation network dataset. In this dataset, the nodes represent researchers. We focused on 246 researchers who have authored 20 or more papers, selected from a corpus of 50,000 papers. The edges (connections) represent co-authorship relationships. There are 1,118 co-authorship connections, where two researchers are connected if they have co-authored one or more papers. We clustered the researchers in this network using both IAC and PLMLE. We set the number of clusters to 8 for simplicity. The results are available at the following link: https://anonymous.4open.science/r/Rebuttal_ICML_8618-9FFF.
> Interestingly, our IAC algorithm found larger communities without excessively small ones, resulting in a more continuous community size distribution. We will include the results and the discussion in the next revision.

---

### Official Review · Reviewer_sjJq · 2025-03-13

**Overall Recommendation:** 4

**Summary:**

The authors propose a new tractable algorithm for cluster recovery in the labeled stochastic block model. An upper bound for the asymptotic error rate is derived and is shown to match known lower bounds.

## Update after rebuttal
I maintain my score, thank you.

**Claims And Evidence:**

The proposed algorithm follows a popular two-phase approach (spectral clustering plus maximum likelihood) and is therefore believable. I did not check the proofs but the arguments as well as the results seem reasonable.

**Essential References Not Discussed:**

N/A

**Experimental Designs Or Analyses:**

Some empirical evaluation is given in the appendix. They seem adequate for this kind of work.

**Methods And Evaluation Criteria:**

See "Claims and evidence" above.

**Other Comments Or Suggestions:**

N/A

**Other Strengths And Weaknesses:**

N/A

**Questions For Authors:**

N/A

**Relation To Broader Scientific Literature:**

The work should make valuable contribution to the literature on community detection and clustering in general.

**Theoretical Claims:**

See "Claims and evidence" above.

---

> ### Author Rebuttal · Authors · 2025-03-31
>
> Thank you very much for your positive feedback and for carefully reading our draft. Please let us know if you have any further questions.

---

### Official Review · Reviewer_a7JY · 2025-03-13

**Overall Recommendation:** 4

**Summary:**

This paper considers the problem of community detection in the Labeled SBM, a generalization of the standard SBM in which each edge is associated with one of L+1 labels (where the zero label is most frequent). The authors study the case of a growing number of communities and propose an algorithm for achieving the minimal misclassification rate, without knowledge of the model parameters. The algorithm consists of two phases: the first performs spectral clustering to identify a preliminary classification and estimate the model parameters, while the second phase refines the initial labels by mimicking the MLE.

**Claims And Evidence:**

Under Assumptions (1), (2), and (3), Theorem 1.2 bounds the misclassification rate. This claim is tight (up to Assumption (3)) in light of Theorem 1.1, which is cited from Yun & Proutiere (2016). The evidence is an algorithm achieving the expected misclassification rate.

**Essential References Not Discussed:**

“Exact recovery and Bregman hard clustering of node-attributed Stochastic Block Model” by Dreveton, Fernandes, and Figueiredo from NeurIPS 2023.
This paper develops very general impossibility results for SBM-type problems.

**Experimental Designs Or Analyses:**

The supplementary material contains synthetic data experiments, showing that the new method performs similarly to the method of Gao et al (2011). It would be nice to see experimental validation of Theorem 1.2.

**Methods And Evaluation Criteria:**

Yes, though it would be nice to see an empirical validation of the threshold in Theorem 1.2.

**Other Comments Or Suggestions:**

There is a grammar issue in the last line of the first paragraph of Section 2.1.

**Other Strengths And Weaknesses:**

The paper is well-written.

**Questions For Authors:**

N/A

**Relation To Broader Scientific Literature:**

There is a good comparison made to the closest papers, namely that of Gao et al (2017) and Yun and Proutiere (2016). The results are significant among the literature on misclassification rates for the SBM.

**Theoretical Claims:**

I checked the outline in the main text, which follows reasonable steps. It makes sense to me that the misclassification rate is driven by the number of vertices that are not in the set H (defined in Section 4.2), since the main condition (H2) is related to the failure of the MLE.

---

> ### Author Rebuttal · Authors · 2025-03-31
>
> Thank you very much for your positive feedback and for carefully reading our draft.
>
> > it would be nice to see an empirical validation of the threshold in Theorem 1.2.
>
> We will include a comparison plot of the empirical error rates and the lower bound by varying $n$ in the appendix of the paper.
>
> > Dreveton, Fernandes, and Figueiredo from NeurIPS 2023. This paper develops very general impossibility results for SBM-type problems.
>
> Thank you for pointing out this literature. This paper derives the necessary conditions for exact recovery with high probability for (possibly non-homogeneous) node-attributed SBMs. Although they provide the Bregman hard clustering algorithm, they do not derive a performance guarantee—that is, the sufficient condition for the existence of the optimal algorithm. Our result provides a guarantee for the IAC algorithm in expectation, and importantly, we have revealed conditions not only for exact recovery but also for the expected number of misclassified nodes to be less than $s$, for any number $s = o(n)$. We will cite their work and discuss it in the revised manuscript.
>
> > There is a grammar issue in the last line of the first paragraph of Section 2.1.
>
> Thank you for pointing this out. We will fix it.

---

> > ### Comment · Reviewer_a7JY · 2025-04-05
> >
> > (copying since this was originally posted as an "official comment")
> > Yes, I agree that the paper of Dreveton et al doesn't cover your results. I brought it up since this paper is not well-known, and thought you would want to be aware of it.

---

> > > ### Author Response · Authors · 2025-04-05
> > >
> > > Thank you again for letting us know—we appreciate you bringing it to our attention!

---

### Official Review · Reviewer_SiLN · 2025-03-16

**Overall Recommendation:** 4

**Summary:**

This paper provides a detailed algorithm for clustering under the LSBM model, and theoretically shows that it achieves the known asymptotic lower bound on the number of miss-classifications (YP2016). Furthermore, the algorithm does not need to know the LSBM parameters, essentially showing that the bound in (YP2016) is min-max over the LSBM parameters. The algorithm is achieved in two stages, where in the first stage a refined version of the spectral clustering method is applied and an estimate of the hyper-parameters is achieved. In the second stage, maximum likelihood principle is applied cyclically to the nodes.

**Claims And Evidence:**

The claims of the paper are mainly theoretical and are rigorously proven. Some empirical evidence is provided in the supplement that confirms the optimality of the proposed algorithm.

**Essential References Not Discussed:**

No undiscussed reference.

**Experimental Designs Or Analyses:**

Experimentation is not the main purpose and approach of this paper.

**Methods And Evaluation Criteria:**

The method provably provides exact bounds for clustering under SBML. It naturally makes sense!

**Other Comments Or Suggestions:**

To me, the section on the related works is unnecessarily long and detailed. The authors could move some of the discussion to the supplementary material and instead present the numerical studies in the main body of the paper.

**Other Strengths And Weaknesses:**

Weakness: I cannot think of any remarkable weakness, but the steps used in the paper are heavily borrowed from other works. The paper is similar even in presentation to (YP2016) and the idea of refinement of the clusters have been frequently discussed in the past, e.g. in K-means clustering, although I am not aware of any paper in the context of LSBMs. The bounds are not also new, but establishing their sharpness is certainly important.

**Questions For Authors:**

No major question to the authors.

**Relation To Broader Scientific Literature:**

The paper establishes the sharpness of the error bounds for LSBM clustering. This is a significant result with potential application in similar problems related to graphs.

**Theoretical Claims:**

I did not check the details of the proofs, but the steps generally follow standard arguments and should be correct.

---

> ### Author Rebuttal · Authors · 2025-03-31
>
> Thank you very much for your positive feedback and for carefully reading our draft.
>
> > The authors could move some of the discussion to the supplementary material and instead present the numerical studies in the main body of the paper.
>
> Thank you for the suggestion. We will take this into consideration when revising the manuscript.

---

### Decision · Program_Chairs · 2025-05-01

**Decision:**

Accept (poster)

**Comment:**

The stochastic block model (SBM) is a natural random-graph model where similar nodes tend to cluster together. The authors study a labeled SBM, where in addition, each edge may be assigned a special label 0 (corresponding to no edge existing); the sparse regime where the probability of a label not being 0 is suitably vanishingly small---O((log n)/n), is considered. In this regime, the authors show that given some assumptions on the LSBM concerning homogeneity of the labels, cluster separation, and the (non)existence of labels that are too sparse, there exists an “instance-optimal algorithm” that will misclassify o(n) of the vertices: necessary and sufficient conditions are developed under which the expected number of misclassified nodes is less than s, for any parameter s = o(n). This algorithm matches a lower bound from the literature. The paper appears to make good progress on a fundamental random-clustering model.